# A methylation-phosphorylation switch controls EZH2 stability and hematopoiesis

**Pengfei Guo, Rebecca C Lim, Keshari Rajawasam, Tiffany Trinh, Hong Sun\*, Hui Zhang\***

Department of Chemistry and Biochemistry, University of Nevada, Las Vegas, Las Vegas, United States

**Abstract** The Polycomb Repressive Complex 2 (PRC2) methylates H3K27 to regulate development and cell fate by transcriptional silencing. Alteration of PRC2 is associated with various cancers. Here, we show that mouse *Kdm1a* deletion causes a dramatic reduction of PRC2 proteins, whereas mouse null mutation of *L3mbtl3* or *Dcaf5* results in PRC2 accumulation and increased H3K27 trimethylation. The catalytic subunit of PRC2, EZH2, is methylated at lysine 20 (K20), promoting EZH2 proteolysis by L3MBTL3 and the CLR4[DCAF5] ubiquitin ligase. KDM1A (LSD1) demethylates the methylated K20 to stabilize EZH2. K20 methylation is inhibited by AKT-mediated phosphorylation of serine 21 in EZH2. Mouse *Ezh2*[K20R/K20R] mutants develop hepatosplenomegaly associated with high GFI1B expression, and *Ezh2*[K20R/K20R] mutant bone marrows expand hematopoietic stem cells and downstream hematopoietic populations. Our studies reveal that EZH2 is regulated by methylation-dependent proteolysis, which is negatively controlled by AKT-mediated S21 phosphorylation to establish a methylation-phosphorylation switch to regulate the PRC2 activity and hematopoiesis.

## Editor's evaluation

This is a valuable study elucidating a novel mechanism of EZH2 regulation. The evidence supporting the claims of the authors is solid, with the inclusion of the large number of data obtained from animal models. This study is of general interest to audiences in the epigenetics field.

\*For correspondence:
hong.sun@unlv.edu (HS);
hui.zhang@unlv.edu (HZ)

**Competing interest:** The authors declare that no competing interests exist.

## Introduction

The PRC2 epigenetically regulates embryonic development and cell fate determination (*Chou et al., 2011*; *Sparmann and van Lohuizen, 2006*). The core components of PRC2 include EZH2 (enhancer of Zeste homolog 2), SUZ12 (suppress of Zeste 12), EED (embryonic ectoderm development), and histone binding proteins RbAp48/46. EZH2 acts as the catalytic subunit of PRC2 to trimethylate histone H3 at lysine 27 (H3K27me3) to promote epigenetic gene silencing during development. Mouse *Ezh2*, *Suz12*, or *Eed* null mutant embryos either cease to develop after implantation or initiate but fail to complete gastrulation (*Faust et al., 1995*; *O'Carroll et al., 2001*; *Pasini et al., 2007*). PRC2 represses developmental regulators in mouse embryonic stem cells and deletion of *Ezh2* compromises the self-renewal and differentiation of human embryonic stem cells by de-repressing developmental regulators (*Collinson et al., 2016*; *Shan et al., 2017*; *Sparmann and van Lohuizen, 2006*). PRC2 is required for various other developmental functions including B-lymphoid development, myogenic differentiation, and imprinted X-chromosome inactivation, and loss of PRC2 also exhausts bone marrow hematopoietic stem cells (*Chou et al., 2011*; *Sparmann and van Lohuizen, 2006*). The pathological role of EZH2 is highlighted by over-expression or gain-of-function mutations in various cancers (*Kim and*

*Roberts, 2016*). EZH2 is considered as an important marker for the aggressive stages of prostate and breast malignancies due to its high expression levels in these cancers (*Kleer et al., 2003*; *Varambally et al., 2002*). EZH2 also serves as a critical therapeutic target of human malignancies including hematopoietic cancers (*Bödör et al., 2013*; *Chang and Hung, 2012*; *Kim and Roberts, 2016*). Many studies indicate that the post-translational modifications of EZH2 play an important role in cancer development (*Li et al., 2020*). One of the critical modifications is the phosphorylation of serine 21 (S21) in EZH2 by AKT, activated by the PI3K signaling cascade (*Cha et al., 2005*). The phosphorylated S21 was reported to inhibit EZH2 methyltransferase activity on H3K27 (*Cha et al., 2005*). Although AKT-mediated S21 phosphorylation on EZH2 is reported to facilitate tumorigenesis (*Kim et al., 2013*), the physiological role and regulation of S21-phosphorylated EZH2 remain largely unclear.

Protein lysine methylation has been extensively investigated in histones to establish the critical roles of mono-, di-, and tri-methylated lysine residues of histones in modulating chromatin structure and gene expression (*Greer and Shi, 2012*; *Zhang et al., 2012*). For example, the methylations of Lys 4 (H3K4), Lys 36 (H3K36), Lys 48 (H3K48), and Lys 79 (H3K79) in histone H3 are typically associated with transcriptional gene activation, but the methylations of Lys 9 (H3K9) and Lys 27 (H3K27) on histone H3, or Lys 20 (H4K20) on histone H4 are usually connected to transcriptional silencing (*Greer and Shi, 2012*). Emerging evidence indicates that many non-histone proteins, such as p53, DNA (cytosine-5)-methyltransferase 1 (DNMT1), NFκB/RelA, ERα, GLI3, SOX2, LIN28A, HIF1α, and E2F1, are mono-methylated on specific lysine residues by SET7 (SET7/9, SET9, SETD7, or KMT7) (*Fu et al., 2016*; *Kim et al., 2014b*; *Lee et al., 2017*; *Leng et al., 2018*; *Zhang et al., 2012*), originally isolated as a histone methyltransferase that mono-methylates H3K4 (*Nishioka et al., 2002*; *Wang et al., 2001*). Our recent studies revealed that specific lysine-methylation on a group of non-histone proteins such as DNMT1, SOX2, SMARCC1, SMARCC2, and E2F1 by SET7 causes lysine methylation-dependent proteolysis of these non-histone proteins (*Guo et al., 2022*; *Leng et al., 2018*; *Wang et al., 2011*; *Zhang et al., 2018*; *Zhang et al., 2019*; *Zhang et al., 2013*). We and others also found that the methyl groups on these methylated non-histone proteins are removed by KDM1A, initially identified as a histone demethylase that specifically removes methyl groups from the mono- and di-methylated H3K4, but not the trimethylated H3K4, to repress transcription (*Guo et al., 2022*; *Lee et al., 2017*; *Leng et al., 2018*; *Shi et al., 2004*; *Wang et al., 2011*; *Yin et al., 2014*; *Zhang et al., 2018*; *Zhang et al., 2013*; *Zhang et al., 2012*).

In this study, we analyzed the phenotypes of *Nestin*-Cre mediated deletion of mouse *Kdm1a* gene. The mouse null mutation of *Kdm1a* causes early embryonic lethality (*Wang et al., 2009*; *Wang et al., 2007*). Loss of mouse *Kdm1a* also profoundly impairs the self-renewal and differentiation of various stem/progenitor cells such as embryonic stem cells and hematopoietic stem cells (*Adamo et al., 2011*; *Saleque et al., 2007*; *Zhang et al., 2018*; *Zhang et al., 2019*). However, the molecular targets of *Kdm1a* deficiency that cause these pathological defects remain largely unclear. We found that loss of *Kdm1a* affects the protein levels of PRC2 and our further analyses reveal that the protein stability of EZH2 is regulated by lysine methylation-dependent proteolysis.

## Results

### Loss of mouse *Kdm1a* in the mouse brain diminishes the levels of PRC2 proteins

We have bred the floxed *Kdm1a* (*Kdm1a*^fl/fl) conditional deletion mice (*Saleque et al., 2007*) with the *Nestin*-Cre transgenic mice (*Tronche et al., 1999*) to specifically delete *Kdm1a* in the central and peripheral nervous system, including neuronal and glial cell precursors. Homozygous loss of *Kdm1a* conditional alleles by breeding with the *Nestin*-Cre mice caused animal lethality immediately after birth on day one (P0, *Figure 1A*). To determine whether the loss of *Kdm1a* affects the PRC2 complex, we examined the levels of PRC2 proteins in the *Kdm1a* null mouse mutants. Immunostaining of brain sections from the wild-type and *Kdm1a* null mutant animals revealed that the protein level of EZH2 is significantly reduced in the *Kdm1a* null mutant mice (*Figure 1A*). Further characterization repeatedly showed that the protein levels of key components of PRC2, including EZH2, SUZ12, and EED, are markedly reduced in the brain extracts of *Kdm1a* null mutants, as compared with those of wild-type littermates (*Figure 1B*). The *Kdm1a* deletion-induced reduction of these PRC2 protein levels occurred post-transcriptionally, since the mRNA levels of the PRC2 key components are comparable between

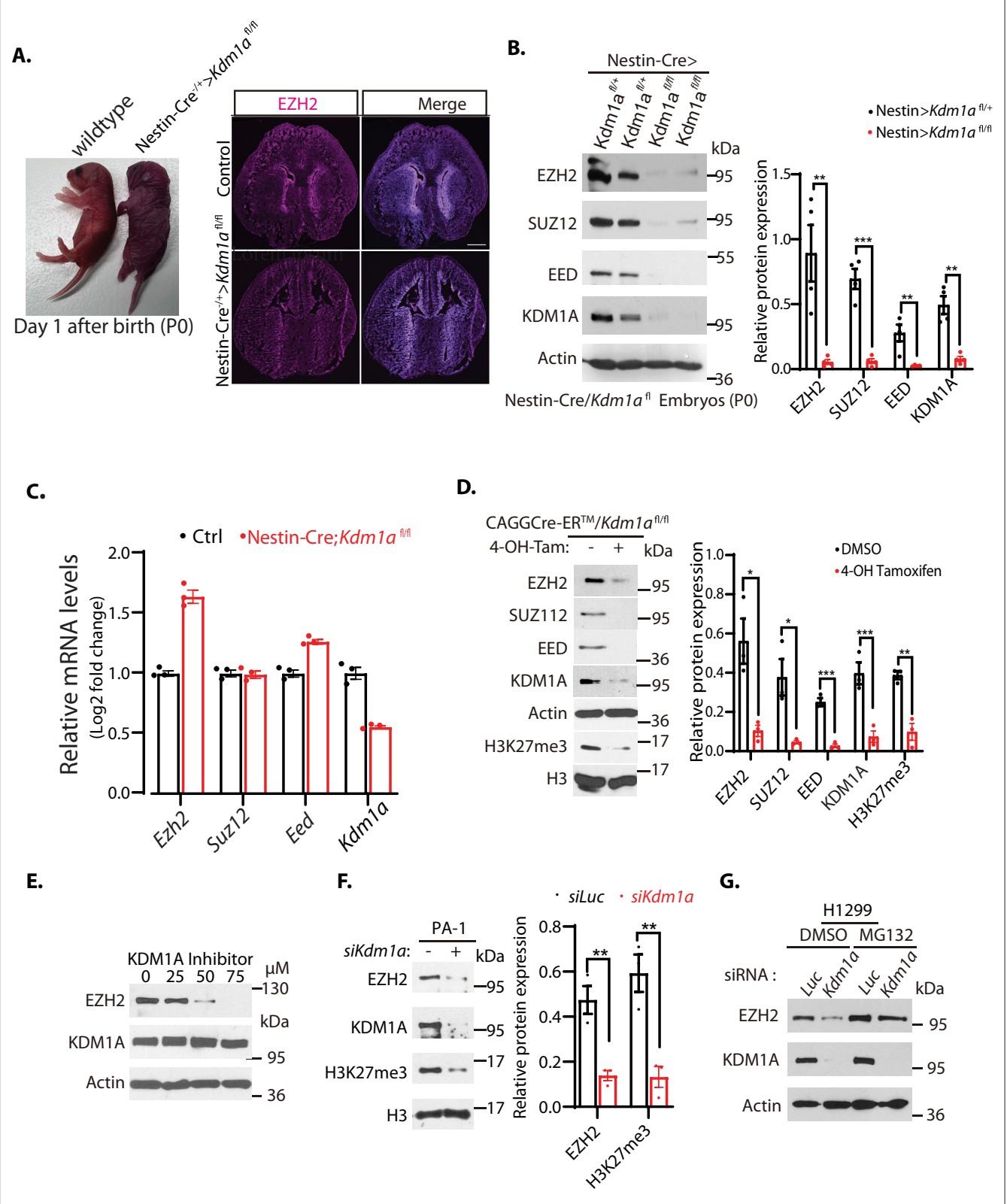

**Figure 1.** Downregulation of EZH2 in *Kdm1a* null mice. (**A**) Left: *Nestin-Cre*-directed conditional inactivation of mouse *Kdm1a* gene causes immediate postnatal death after birth (**P0**). Right: Immunofluorescence staining of EZH2 from the brain sections of the wild-type and *Nestin-Cre;Kdm1a* fl/fl mice on P0 day. Scale bars, 500 μm. (**B**) Polycomb repressive complex 2 (PRC2) protein levels in the brain extracts of P0 *Nestin-Cre;Kdm1a* fl/+ heterozygous control or *Nestin-Cre;Kdm1a* fl/fl homozygous *Kdm1a* conditional deletion mice were analyzed by Western blotting. (**C**) Reverse-transcriptional

*Figure 1 continued on next page*

*Figure 1 continued*

quantitative PCR (RT-qPCR) analysis of the mRNA levels of *Ezh2*, *Suz12*, *Eed,* and *Kdm1a* in the brain of the *Nestin-Cre;Kdm1a* $^{fl/fl}$ mice. The mRNA levels were measured in triplicate by RT-qPCR. Quantifications are represented by bar graph with mean and standard deviation (S.D.) for error bars from three replicate samples and normalized to the control wild-type *Kdm1a* $^{fl/fl}$ mice. (**D**) Excision of *Kdm1a* by 4-OH-TAM in the *Kdm1a* $^{fl/fl}$-Actin-Cre $^{ER}$ MEFs reduces PRC2 proteins. Embryonic fibroblasts from *CAGGCre-ER/Kdm1a* $^{fl/fl}$ mouse embryos (E13.5) were treated with 4-hydroxytamoxifen (20 µg/ ml) for 12 hr to delete *Kdm1a* by inducible Actin-Cre-ER. (**E**) Wild-type MEFs were treated with various concentrations of KDM1A inhibitor CBB3001 (20 µM) for 20 hr and EZH2 and KDM1A protein levels were analyzed by blotting with indicated antibodies. (**F**) PA-1 cells were transfected with 50 nM luciferase control (Luc) or *Kdm1a* (*Kdm1a*-1) siRNAs for 48 hr and the levels of indicated proteins were analyzed. (**G**) H1299 cells were transfected with 50 nM luciferase (Luc) control or *Kdm1a* (*Kdm1a-1*) siRNAs for 48 hr and added 5 µg/ml MG132 for the last 6 hr before lysing the cells for blotting. For (**B**), (**D**) and (**F**), Significance was indicated as a two-tailed, unpaired, *t*-test. Values are expressed as the mean ± SEM. *p<0.05. **p<0.01. ***p<0.001. Protein molecular weight markers are in kilodalton (kDa).

The online version of this article includes the following source data and figure supplement(s) for figure 1:

**Source data 1.** Original photos for *Figure 1A* and original blots for *Figure 1B–G*.

**Source data 2.** Original table sources for quantification of *Figure 1* plots.

**Figure supplement 1.** Regulation of EZH2 by KDM1A.

**Figure supplement 1—source data 1.** Original blots for *Figure 1—figure supplement 1*.

the *Kdm1a* null mutants and the wild-type littermates in the brain tissues (*Figure 1C*). To further examine if the reduction of PRC2 proteins is caused by animal lethality, we established mouse embryonic fibroblasts (MEFs) from the homozygous *Kdm1a* $^{fl/fl}$ deletion mouse embryos with the actin-Cre-ER (CAGGCre-ER/*Kdm1a* $^{fl/fl}$) by breeding the *Kdm1a* $^{fl/fl}$ conditional deletion mouse strain (*Saleque et al., 2007*) with a transgenic mouse strain expressing a tamoxifen-inducible Cre-ER recombinase under the actin promoter control (CAGGCre-ER) (*Hayashi and McMahon, 2002*). While the wild-type MEFs normally express substantial levels of PRC2 proteins, induced deletion of *Kdm1a* in the CAGGCre-ER/*Kdm1a* $^{fl/fl}$ MEFs by addition of 4-hydroxytamoxifen (4-OH-Tam) led to the rapid disappearance of these PRC2 proteins and reduction of H3K27me3 (*Figure 1D*). Since the loss of a single core component of PRC2, such as EZH2 or SUZ12, typically leads to the disassembly of PRC2 and proteolysis of other PRC2 subunits (*Collinson et al., 2016*; *Pasini et al., 2004*), we further determined if the stability of PRC2 proteins such as EZH2 is dependent on *Kdm1a* deletion. We treated the CAGGCre-ER/*Kdm1a* $^{fl/fl}$ MEFs with a KDM1A inhibitor, CBB3001, that we previously developed (*Guo et al., 2022*; *Hoang et al., 2018*) and found that the protein level of EZH2 is significantly decreased by CBB3001 (*Figure 1E*). We also used siRNA-mediated silencing of *Kdm1a* in cultured human terato-carcinoma PA-1 cells (*Figure 1F*), lung carcinoma H1299 cells (*Figure 1G*), and lung carcinoma H520 cells (*Figure 1—figure supplement 1A*). We found that *Kdm1a* silencing caused the downregulation of EZH2 protein and the *Kdm1a* silencing-induced EZH2 reduction is reversed by the treatment of 26 S proteasome inhibitor, MG132, added in the last 6 hr of the experiment (*Figure 1G*; *Guo et al., 2022*). To exclude the potential off-target effect of *Kdm1a* siRNAs, we stably expressed Flag-tagged wild-type *Kdm1a* cDNA that does not contain the 3'*UTR* region of the endogenous *Kdm1a* gene or the catalytically dead mutant that contains only the amino-terminal 1–531 amino acid residues of the Flag-tagged *Kdm1a* cDNA in H1299 cells. We found that while the siRNA against the 3'*UTR* region (si-*Kdm1a*-3'*UTR*) of the endogenous *Kdm1a* leads to the downregulation of EZH2 in the absence of the Flag-KDM1A, expression of the wild-type, but not the catalytically dead mutant, can suppress the si-*Kdm1a*-3'*UTR* effects of the endogenous *Kdm1a* on EZH2 proteins (*Figure 1—figure supplement 1B and C*). Our studies indicate that KDM1A is required to maintain the protein stability of EZH2 to prevent the disassembly of PRC2.

## L3MBTL3 regulates the stability of EZH2 protein

Since KDM1A serves as a demethylase for the mono- and di-methylated histone H3K4 and several lysine-methylated non-histone proteins, including DNMT1, E2F1, SOX2, SMARCC1, and SMACC2 (*Guo et al., 2022*; *Leng et al., 2018*; *Zhang et al., 2018*; *Zhang et al., 2019*), we wondered whether PRC2 proteins are regulated by the lysine methylation-dependent proteolysis pathway through L3MBTL3, a methyl lysine reader that binds to the mono-methylated DNMT1, SOX2, and SMARCC1 (*Leng et al., 2018*; *Zhang et al., 2019*). Our initial study is to examine and characterize EZH2 and SUZ12 proteins in *L3mbtl3* null mouse embryos (*Guo et al., 2022*). We repeatedly found that the levels of EZH2 and SUZ12 proteins are significantly increased in the mouse *L3mbtl3* null embryos,

which died at E17.5–19.5, as compared to that of the wild-type littermates (*Figure 2A*). In mouse embryonic fibroblasts isolated from the wild-type or *L3mbtl3* homozygous deletion mutant embryos, the loss of *L3mbtl3* caused the accumulation of EZH2 protein and the increased level of trimethylated H3K27 (*Figure 2B*).

Since the homozygous loss of mouse *L3mbtl3* impairs the maturation of the mouse hematopoietic system to cause anemia and embryonic lethality around E17.5-E19.5 (*Arai and Miyazaki, 2005*), we employed mouse *L3mbtl3*^tm1a(EUCOMM)Hmgu embryonic stem cells to generate the conditional floxed *L3mbtl3*^fl/fl mice by removing the neo-LacZ elements with the Flp recombinase to establish the loxP sites that flank the exon 5 of the *L3mbtl3* allele in the mice (*Figure 2—figure supplement 1*). To induce the homozygous *L3mbtl3*^fl/fl conditional deletion in the central nervous system, the *L3mbtl3*^fl/fl mice were bred with *Nestin*-Cre transgenic mice. We found that homozygous loss of *L3mbtl3* in the brain of the *L3mbtl3*^fl/fl/*Nestin*-Cre mice survived, but the brain extracts accumulated EZH2 and SUZ12 proteins, as compared with that of *L3mbtl3*^fl/+/*Nestin*-Cre mice (*Figure 2C*). Our immunostaining of the brain sections from the wild-type and *L3mbtl3*^fl/fl/*Nestin*-Cre null mutant animals also revealed that the protein levels of EZH2 and trimethylated H3K27 accumulate in the *L3mbtl3* conditional null mutant brains, as compared to that of the control wild-type littermates (*Figure 2D*). To further confirm that EZH2 is regulated by KDM1A and L3MBTL3 in vivo, the *L3mbtl3* conditional deletion mouse strain were bred with the *Kdm1a*^fl/fl conditional deletion strain and the transgenic mouse expressing a tamoxifen-inducible Cre-ER recombinase under the actin promoter control (CAGGCre-ER) to generate the inducible conditional knock-out mice of either *Kdm1a* or *L3mbtl3* alone, or the double conditional knock-out mice of both *Kdm1a* and *L3mbtl3*. The MEFs were established from the homozygous CAGGCre-ER/*Kdm1a*^fl/fl and the double CAGGCre-ER/ *Kdm1a*^fl/fl/*L3mbtl3*^fl/fl deletion embryos for *Kdm1a* and *Kdm1a/L3mbtl3* deletions. While these MEFs normally express substantial levels of EZH2 and H3K27me3, induced deletion of *Kdm1a* in the CAGGCre-ER/*Kdm1a*^fl/fl MEFs by addition of 4-hydroxytamoxifen (4-OH-Tam) led to the disappearance of EZH2 protein and the corresponding reduction of H3K27me3 (*Figure 2E*). However, the induced co-deletion of *Kdm1a* and *L3mbtl3* in the CAGGCre-ER/ *Kdm1a*^fl/fl/*L3mbtl3*^fl/fl MEFs leads to the restoration of the EZH2 and H3K27me3 protein levels in the MEFs. We also used siRNA-mediated silencing of *L3mbtl3* in human lung carcinoma H1299 cells with two representative siRNAs (*Guo et al., 2022*; *Leng et al., 2018*). Our study revealed that while *Kdm1a* silencing reduced EZH2 protein levels, co-silencing of *L3mbtl3* and *Kdm1a* stabilized EZH2 protein levels in *Kdm1a*-deficient cells (*Figure 2F and G*).

Since siRNA-mediated silencing usually reduces a fraction of target gene expression in cultured cells, we further used the CRISPR-Cas9 gene editing to make homozygous deletion of the *L3mbtl3* gene in the HCT116 cell line (*Doench et al., 2014*; *Sanjana et al., 2014*; *Shalem et al., 2014*). Analysis of EZH2 protein revealed that EZH2 protein levels are elevated in the CRISPR-Cas9-mediated *L3mbtl3* knockout (KO) cells than the parental wild-type HCT116 cells (*Figure 2—figure supplement 2A*). To further evaluate the effect of L3MBTL3 on EZH2, we also ectopically and stably expressed the Flag-tagged L3MBTL3 in HCT116 cells. Our studies showed that L3MBTL3 overexpression in HCT116 cells caused the downregulation of EZH2 protein (*Figure 2—figure supplement 2B*). Collectively, these studies indicate the protein level of EZH2 is tightly regulated by L3MBTL3.

## DCAF5 controls EZH2 and H3K27me3 levels

Our previous studies have shown that L3MBTL3 recruits the CRL4^DCAF5 ubiquitin E3 ligase complex to target substrates, such as DNMT1, SOX2, and SMARCC1, for ubiquitin-dependent proteolysis (*Guo et al., 2022*; *Leng et al., 2018*; *Zhang et al., 2019*). To determine whether DCAF5, a substrate-specific subunit of the CRL4 ubiquitin E3 ligase complex (*Leng et al., 2018*), is involved in regulating the levels of EZH2 and H3K27me3, we employed the CRISPR-Cas9 gene editing system to delete the exon 4 of the mouse *Dcaf5* allele to establish the *Dcaf5* deletion mutant mouse strain with a stop codon to the remaining downstream reading frame of the *Dcaf5* allele (*Figure 3A and B*; *Kim et al., 2014a*; *Kleinstiver et al., 2016*; *Slaymaker et al., 2016*). We found that homozygous mutation of the *Dcaf5* alleles also caused significant elevation of EZH2 and H3K27me3 protein levels (*Figure 3C*). Immunostaining of the embryonic brain sections revealed that loss of *Dcaf5* leads to increased levels of EZH2 and H3K27me3 staining, as compared with that of control wild-type littermates (*Figure 3D*). Consistent with these animal studies, siRNA-mediated co-silencing of *Dcaf5* and *Kdm1a* in H1299 cells with two representative siRNAs stabilized EZH2 protein in *Kdm1a*-deficient cells (*Figure 3E and*

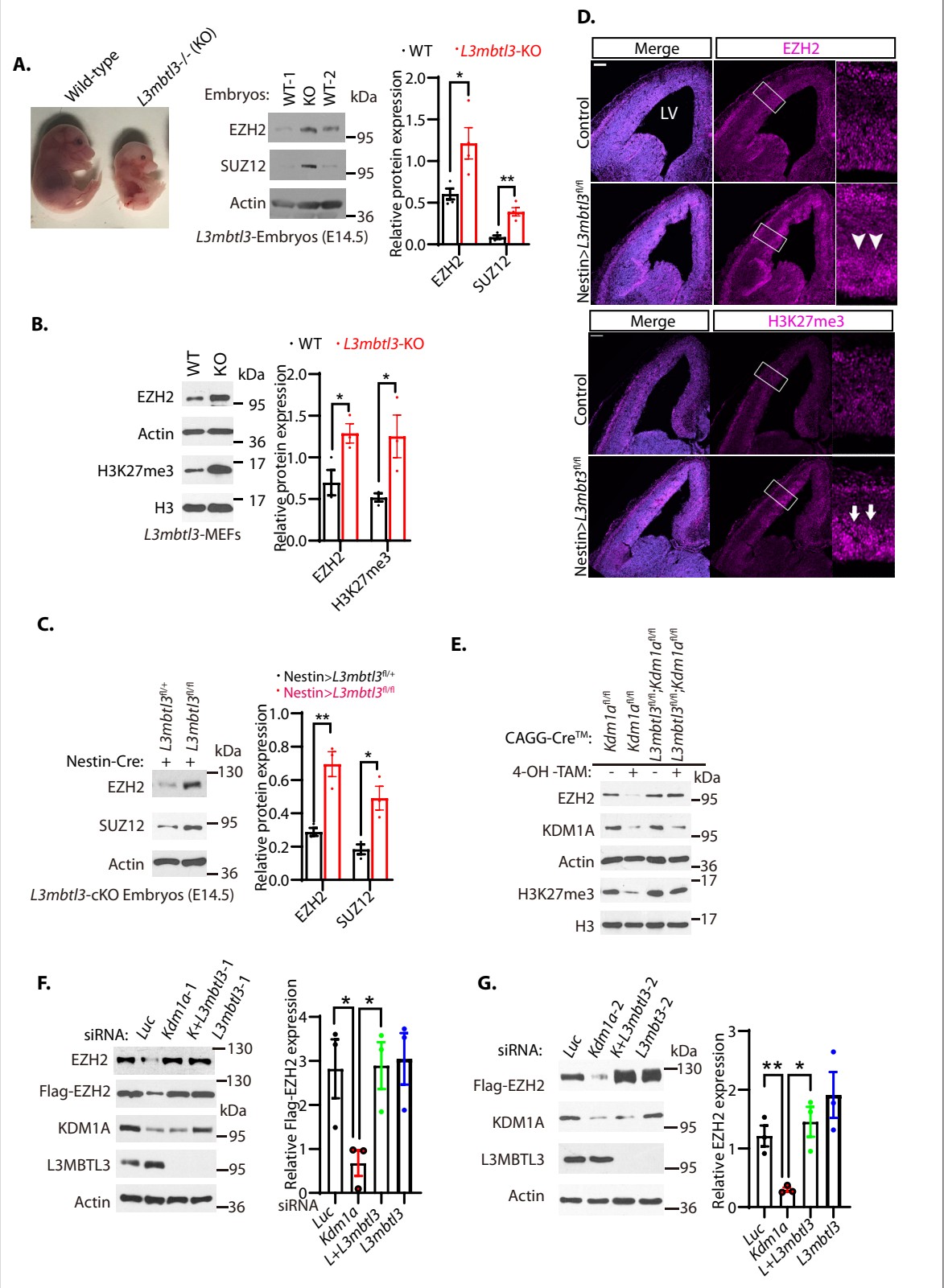

**Figure 2.** Loss of L3MBTL3 stabilizes EZH2 protein. (**A**) Left: The mouse *L3mbt3* wild-type (+/+) and *L3mbtl3* null (-/-, KO) mutant embryos on embryonic day 17.5 (E17.5) after breeding. Right: Total lysates from the heads of mouse *L3mbt3 (+/+)* wild-type and *L3mbt3* homozygous deletion (-/-, KO) mutant embryos (equal total proteins) were analyzed by Western blotting with antibodies for the indicated proteins. (**B**) Mouse embryonic fibroblasts from the wild-type and *L3mbtl3* deletion mutant embryos (E13.5) were examined for EZH2 and H3K27me3 proteins by Western blotting. (**C**) Western blot analysis

*Figure 2 continued on next page*

*Figure 2 continued*

of EZH2 and SUZ12 proteins in the head extracts of Nestin-Cre;L3mbtl3^fl/+ control and *Nestin-Cre;L3mbtl3^fl/fl* conditional deletion (cKO) embryos (E14.5) using the indicative antibodies. (**D**) Coronal sections of the developing mouse brain at E15.5 were stained with anti-EZH2 and H3K27me3 antibodies in wild-type control and *Nestin-Cre;L3mbtl3^fl/fl* conditional deletion mice. Scale bars, 100 μm. Arrows and arrowheads indicate the regions of EZH2 and H3K27me3 expression, respectively. LV: lateral ventricle. (**E**) Deletion of both *Kdm1a* and *L3mbtl3* by 4-OH-TAM restores the protein levels of EZH2 and H3K27me3. MEFs from the *Nestin-Cre;Kdm1a^fl/fl* and *CAGGCre-ER;Kdm1a^fl/fl;L3mbtl3^fl/fl* mouse embryos (E13.5) were treated with 4-hydroxytamoxifen (4-OH-TAM, 20 μg/ml) for 12 hr to delete *Kdm1a* and *L3mbtl3*. (**F**) Silencing of *L3mbtl3* re-stabilizes the protein levels of EZH2 in *Kdm1a* deficient cells. The Flag-EZH2 under the retroviral LTR promoter control were ectopically and stably expressed in H1299 cells and the cells were transfected with 50 nM siRNAs of luciferase (Luc), *Kdm1a* (*Kdm1a*-1), and *L3mbtl3* (*L3mbtl-1*) siRNAs. (**G**) Silencing of *L3mbtl3* stabilizes EZH2 protein in *Kdm1a* deficient cells. The Flag-EZH2 under the retroviral LTR promoter control were ectopically and stably expressed in H1299 cells and the cells were transfected with 50 nM siRNAs of luciferase (*Luc*), *Kdm1a* (*Kdm1a-2*), *Kdm1a* and *L3mbtl3* (*L3mbtl3-2*), and *L3mbtl3* siRNAs. The indicated proteins were analyzed by Western blotting. For (**A–C**), (**F**), and (**G**), band intensities were quantified and normalized to that of the luciferase or actin control. Significance was indicated as a two-tailed, unpaired, *t*-test. Values are expressed as the mean ± SEM. *p<0.05. **p<0.01. ***p<0.001.

The online version of this article includes the following source data and figure supplement(s) for figure 2:

**Source data 1.** Original blots for *Figure 2A–C*, original photos for *Figure 2D*, and original blots for *Figure 2E–G*.

**Source data 2.** Original table sources for quantification of *Figure 2* plots.

**Figure supplement 1.** Schematic illustration to generate a conditional flox allele in *L3mbtl3^fl/fl* with the *FRT-loxP* sites at the exon 5 of the *L3mbtl3* locus.

**Figure supplement 2.** The EZH2 protein is regulated by L3MBTL3.

**Figure supplement 2—source data 1.** Original blots for *Figure 2—figure supplement 2*.

*F*). These results indicate that the CRL4^DCAF5 ubiquitin ligase complex is involved in the proteolytic degradation of EZH2 to regulate the H3K27me3 levels during mouse development and in cultured cancer cells.

## The protein stability of EZH2 is regulated by lysine methylation

Our mouse genetic evidence indicates that the protein stability of EZH2 is regulated by KDM1A, L3MBTL3, and DCAF5, which are involved in regulating the proteolytic degradation of lysine methyl-ated protein substrates, such as DNMT1, SOX2, SMARCC1, and SMARCC2 (*Guo et al., 2022*; *Leng et al., 2018*; *Zhang et al., 2018*; *Zhang et al., 2019*). To further investigate EZH2 regulation, we examined and found that EZH2 contains a conserved lysine residue, Lys20 (K20), in the putative SET7 methylation consensus motif (R/K-S/T/V-K) that is very similar to that of the Lys42 (K42) methylation degron motif in SOX2 (*Figure 4A*; *Zhang et al., 2018*; *Zhang et al., 2019*). To test whether the K20 residue in EZH2 serves as a putative substrate for KDM1A, we synthesized the monomethylated K20 peptide and its unmethylated cognate peptide derived from EZH2 (*Figure 4—figure supplement 1*). To effectively detect the methylated K20 in EZH2, we generated and affinity purified a specific anti-monomethylated K20 (K20me) peptide antibody for EZH2, which recognized the methylated K20 peptides containing the mono-methylated K20 (K20me) peptides, but not the unmethylated cognate K20 peptide or the monomethylated or cognate unmethylated K17 peptide (*Figure 4—figure supplement 1*). We mixed and incubated the monomethylated K20 peptide with purified GST-KDM1A and found that the recombinant GST-KDM1A protein can effectively remove the methyl group of the monomethylated K20 peptide (*Figure 4C*), indicating that the monomethylated K20 in EZH2 serves as a direct substrate of KDM1A. To validate whether K20 of EZH2 can be methylated in vivo, we measured the methylation levels of EZH2 using the anti-EZH2-K20me antibody in *Kdm1a* siRNA-mediated knockdown of HCT116 cells. Loss of KDM1A destabilizes both EZH2 and EZH2-K20me and reduces the level of H3K27me3 in HCT116 cells, but treatment of KDM1A-deficient cells with the protease inhibitor MG132 restored the levels of EZH2, EZH2-K20me, and H3K27me3 (*Figure 4D*). We also measured the EZH2-K20me levels in *Kdm1a*^fl/fl conditional and *L3mbtl3*-knockout mice. As shown in *Figure 4E*, *Nestin*-Cre-mediated loss of *Kdm1a* in the mouse caused the reduction of both the K20 methylation level of EZH2 and the EZH2 protein levels, as well as the downregulation of trimethylated H3K27 levels. Conversely, homozygous loss of *L3mbtl3* resulted in an increased level of EZH2 K20me (*Figure 4F*).

We next assessed whether the SET7 methyltransferase is involved in the L3MBTL3-mediated binding to the methylated EZH2 in HCT116 cells. We found that transient overexpression of SET7 resulted in a reduction of EZH2 protein and H3K27 trimethylation levels in HCT116 cells (*Figure 4G*). In addition, the K20-methylated EZH2 fraction is enriched in the L3MBTL3 co-immunoprecipitation

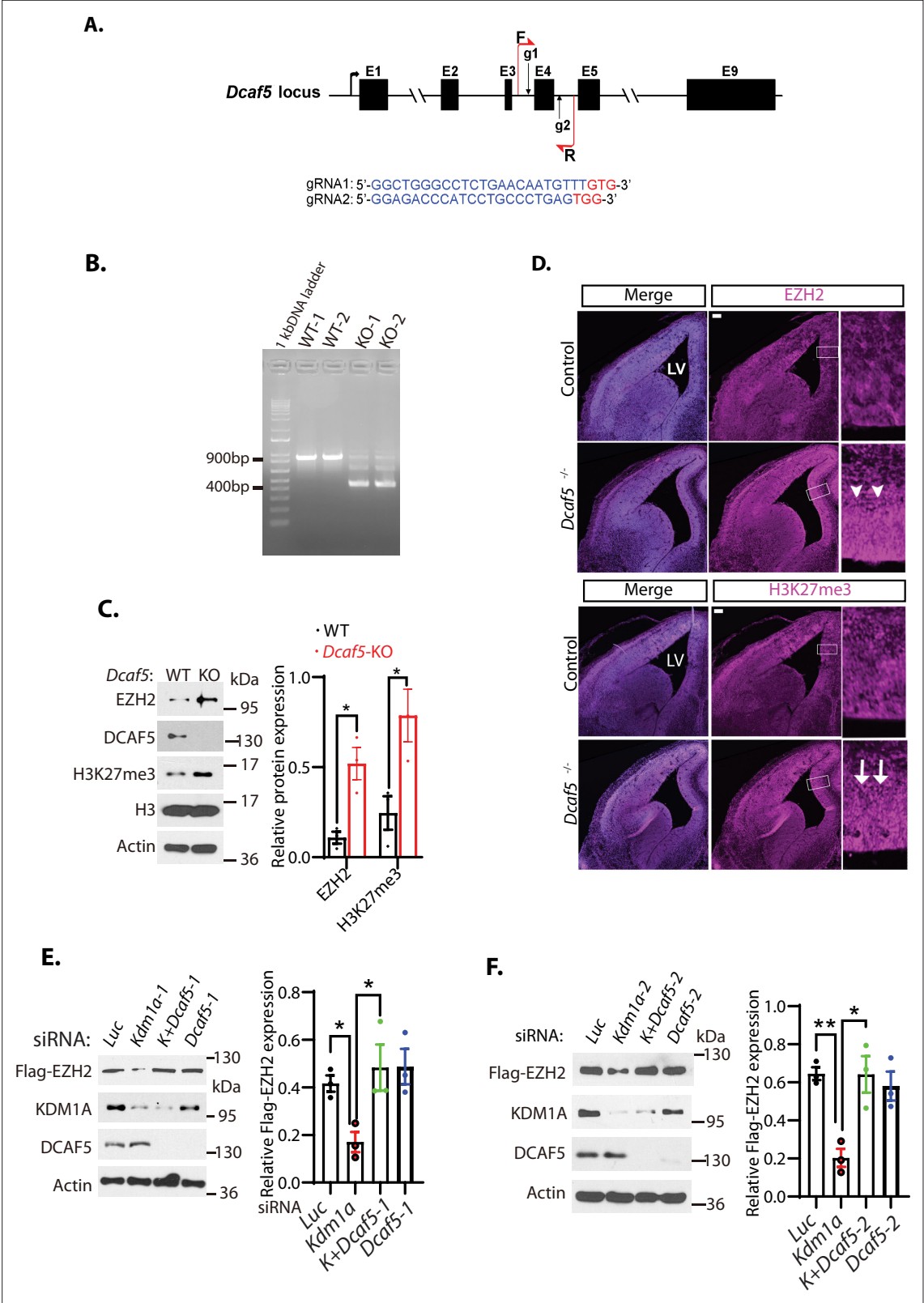

**Figure 3.** Loss of *Dcaf5* stabilizes EZH2 protein. (**A**) The strategy to delete the exon 4 of the mouse *Dcaf5* gene by CRISPR-Cas9 gene edition with two guide RNAs (gRNAs). (**B**) Genome typing of the wild-type (WT) and *Dcaf5* knock-out (KO) mice by PCR. (**C**) Western blot analysis of EZH2 and H3K27me3 proteins in the brains of the wild-type control and homozygous *Dcaf5* deletion mice using the indicative antibodies. (**D**) Accumulation of EZH2 and H3K27me3 proteins in mouse *Dcaf5* deleted embryonic brains. Immunostainings of anti-EZH2 and H3K27me3 in the coronal sections of the

*Figure 3 continued on next page*

*Figure 3 continued*

mouse embryonic brains of the wild-type control and *Dcaf5* at E15.5. Scale bars, 100 μm. Arrows and arrowheads indicate the expression regions of EZH2 and H3K27me3, respectively. Boxed regions are enlarged on the right panels. LV: lateral ventricle. (**E**) Silencing of *Dcaf5* re-stabilizes the protein levels of EZH2 in *Kdm1a* deficient cells. H1299 cells expressing stably expressed Flag-EZH2 were transfected with 50 nM siRNAs of luciferase (Luc), *Kdm1a-1*, *Kdm1a* and *Dcaf5-1*, and *Dcaf5-1* siRNAs. The indicated proteins were analyzed by Western blotting. (**F**) Silencing of *Dcaf5* re-stabilizes the protein levels of EZH2 in *Kdm1a* deficient cells. H1299 cells expressing stably expressed Flag-EZH2 were transfected with 50 nM siRNAs of luciferase (Luc), *Kdm1a-2*, *Kdm1a-2+Dcaf5-2*, and *Dcaf5-2* siRNAs. The indicated proteins were analyzed by Western blotting. Band intensities in (**C**), (**E**), and (**F**) were quantified and normalized to that of the histone H3 or luciferase control. Significance was indicated as a two-tailed, unpaired, *t*-test. Values are expressed as the mean ± SEM. *p<0.05. **p<0.01.

The online version of this article includes the following source data for figure 3:

**Source data 1.** Original photos for *Figure 3B and D*, original blots for *Figure 3C, E and F*.

**Source data 2.** Original table sources for quantification of *Figure 3* plots.

with EZH2 in SET7-expressed and MG132-treated cells (*Figure 4H*). Although SET7 is only transiently expressed in these cells (*Figure 4G and H*), our studies indicate that the fractions of cells that express SET7 can promote the interaction between EZH2 and L3MBTL3, as well as EZH2 proteolysis. Our investigation further showed that while silencing of *Kdm1a* reduced the protein level of ectopically and stably expressed Flag-tagged EZH2 in H1299 cells, co-silencing of *Set7* with *Kdm1a* siRNAs effectively re-stabilized Flag-EZH2 protein in KDM1A-deficient cells (*Figure 4I* and *Figure 4—figure supplement 2A*). These studies indicate that SET7 is capable of methylating K20 in EZH2, and that the methylated K20 of EZH2 is targeted by L3MBTL3 for methylation-dependent proteolysis of EZH2.

Since the K20 protein methylation motif of EZH2 is similar to the methylation degron motif of K42 in SOX2 (*Figure 4A*), and our previous studies showed that the methylated K42 of SOX2 is recognized by PHF20L1 to prevent the degradation of SOX2 (*Zhang et al., 2018*; *Zhang et al., 2019*), we examined whether loss of PHF20L1 causes the proteolysis of EZH2. We found that silencing of *Phf20l1* by two independent siRNAs both induced the degradation of EZH2 protein, and co-silencing of *Phf20l1* and *L3mbtl3* restored EZH2 protein levels (*Figure 4—figure supplement 2B and C*). These studies indicate that PHF20L1 normally prevents the proteolysis of EZH2 by the methylation-dependent proteolysis.

Using the affinity-purified K20me antibodies, we examined K20 methylation of EZH2 in human breast ductal carcinoma T47D cells, as this cell line was previously used to analyze the effect of AKT-mediated serine phosphorylation in EZH2 (*Cha et al., 2005*). We found that ectopic expression of SET7 led to substantially increased K20 methylation of EZH2 and H3K27me3 proteins (*Figure 4—figure supplement 2D*). However, while SET7 expression reduced EZH2 protein in HCT116 cells (*Figure 4G*), expression of SET7 in T47D cells did not cause significant downregulation of EZH2 protein, although the levels of K20 methylation is increased after SET7 is expressed (*Figure 4—figure supplement 2D*). On the other hand, SET7 expression and increased K20 methylation of EZH2 in T47D cells led to a substantial reduction of the phosphorylated serine 21 (S21) in EZH2, a site previously reported to be phosphorylated by the active AKT activity (*Cha et al., 2005*). We noticed that the K20 residue is immediately next to S21 in EZH2 (*Figure 4A and B*, and *Figure 4—figure supplement 1*). Previous studies have shown that S21 of EZH2 is phosphorylated by the activated AKT to suppress the activity of EZH2 for H3K27 trimethylation, but S21 phosphorylation does not alter EZH2 protein stability in T47D cells (*Cha et al., 2005*). Our results from T47D cells indicate that transient expression of SET7 catalyzes an increased level of the mono-methylation of K20 and also causes the decrease of the phosphorylation of S21 in EZH2 in T47D cells (*Figure 4—figure supplement 2D*), suggesting that the methylation of K20 may negatively regulate the phosphorylation of S21 of EZH2. However, we propose to further investigate why increased K20-methylation of EZH2 by SET7 expression does not cause significant proteolysis of EZH2 in T47D cells (see below).

To determine the regulation of EZH2 by K20 methylation and S21 phosphorylation, we analyzed their levels during mouse embryonic development. Our studies revealed that the methylated K20 levels of EZH2 gradually increased during mouse embryogenesis from E14.5, E18.5, to the first postnatal day after birth (P0), associated with the gradual downregulation of EZH2 protein and H3K27 trimethylation levels at these development stages (*Figure 5A*). The downregulation of the trimethylated H3K27 is also accompanied by the increased levels of S21-phosphorylated form of EZH2 during the indicated developmental stages, suggesting that the activity of EZH2 is reduced by

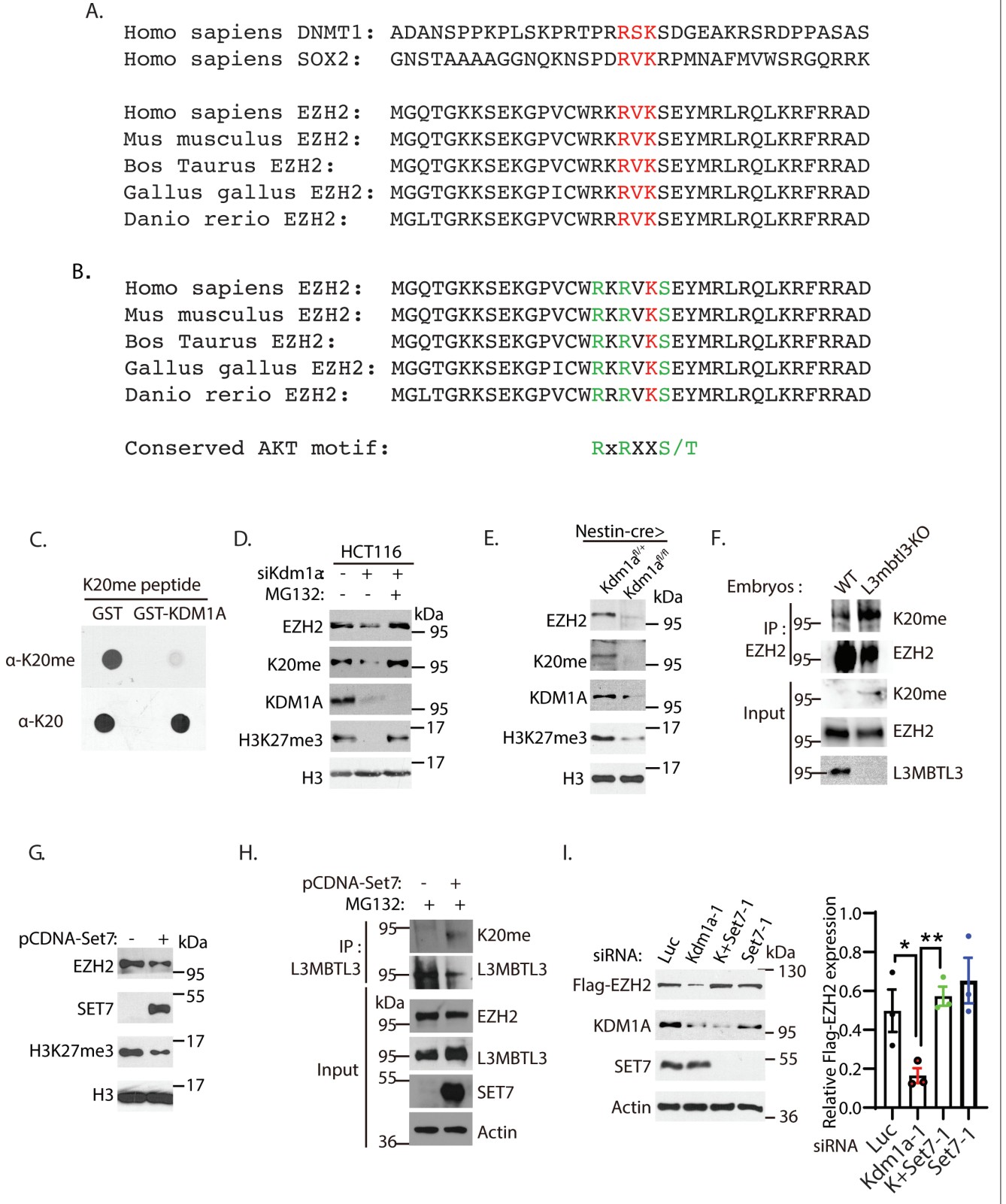

**Figure 4.** The K20 residue in EZH2 is methylated by SET7. (**A**) EZH2 contains a conserved lysine residue, Lys 20 (**K20**), within the consensus lysine residues (K*) of the H3K4-like R/K-S/T/V-K* methylation motifs methylated by SET7. (**B**) K20 (red) is next to serine 21 (**S21**) that is phosphorylated by AKT in EZH2. The critical amino acid residues (RXRXXS) in the AKT phosphorylation consensus motif are labeled green. (**C**) KDM1A demethylates the methylated K20 peptide. Purified 1 μg GST control or GST-KDM1A proteins were incubated with 100 ng of mono-methylated K20 peptides for 4 hr

*Figure 4 continued on next page*

*Figure 4 continued*

at room temperature and the resulting peptides and input methylated peptides were spotted onto nitrocellulose membrane and blotted with affinity purified anti-methylated K20 and anti-K20 antibodies as indicated. (**D**) HCT116 cells were transfected with 50 nM luciferase (*Luc*) control or *Kdm1a* siRNAs for 48 hr and added 5 µg/ml MG132 or DMSO for the last 6 hr before lysing the cells for blotting. Proteins were detected by Western blotting with indicative antibodies. (**E**) The protein levels of EZH2, EZH2-K20me, and H3K27me3 in the brain extracts of E18.5 *Nestin-Cre;Kdm1a*[fl/+] heterozygous control or *Nestin-Cre;Kdm1a*[fl/fl] homozygous *Kdm1a* conditional deletion mice were analyzed by Western blotting. (**F**) The *L3mbt3* wild-type and deletion (−/−) mutant embryos on embryonic day 15.5 (**E15.5**) were analyzed for monomethylated K20 of EZH2. Total lysates were immunoprecipitated with EZH2 antibody and blotted with indicated antibodies. (**G**) HCT116 cells were transfected with control vector (pcDNA3) or SET7 expression construct for 48 hr and protein extracts were prepared. Proteins were detected by Western blotting with indicative antibodies (**H**) The K20-methylated EZH2 preferentially binds to L3MBTL3. The 293T cells were transfected with control vector (pcDNA3) or SET7 expression construct for 48 hr, and proteasome inhibitor MG132 (5 µg/ml) was added for the last 6 hr. Interactions between L3MBTL3 and EZH2-K20me were analyzed by co-immunoprecipitation and Western blotting analyses. (**I**) Silencing of SET7 re-stabilizes the protein levels of EZH2 in KDM1A deficient cells. H1299 cells expressing stably expressed Flag-EZH2 were transfected with 50 nM siRNAs of luciferase (*Luc*), *Kdm1a* (*Kdm1a-1*), *Set7* (*Set7-1*) siRNAs and their combination. The indicated proteins were analyzed by Western blotting. The protein bands were quantified and normalized to that of the luciferase control. Significance was indicated as a two-tailed, unpaired, *t*-test. Values are expressed as the mean ± SEM. *p<0.05. **p<0.01.

The online version of this article includes the following source data and figure supplement(s) for figure 4:

**Source data 1.** Original blots for *Figure 4C–I*.

**Source data 2.** Original table sources for quantification of *Figure 4* plots.

**Figure supplement 1.** Specificity of anti-methylated K20 peptide antibodies.

**Figure supplement 1—source data 1.** Original dot blots for *Figure 4—figure supplement 1*.

**Figure supplement 2.** Loss of *L3mbtl3* reduces the decrease of EZH2 protein induced by *Kdm1a* silencing.

**Figure supplement 2—source data 1.** Original blots for *Figure 4—figure supplement 2*.

---

both K20-methylation-dependent proteolysis and S21-phosphorylation dependent activity inhibition (*Figure 5A*). To verify that L3MBTL3 is expressed during mouse development to target EZH2 for proteolysis, we established the mouse *L3mbtl3* promoter-regulated LacZ expression in the mice by breeding the *L3mbtl3*[fl/fl] mouse with *Sox2*-Cre, which removed the neo-flox element in the exon 4 of the *L3mbtl3* allele from the mice to express LacZ under the *L3mblt3* promoter control (*Figure 2—figure supplement 1*). The staining of *L3mbtl3*-LacZ in the developing mouse embryos revealed that *L3mbtl3*-LacZ is expressed around E12-E15 during mouse development (*Figure 5B*), with the highly elevated expression of L3MBTL3 around embryonic day E15. These studies suggest that L3MBTL3 is expressed during the mouse embryonic stages when the protein levels of EZH2 and H3K27me3 start to decline.

In mouse embryonic extracts, our co-immunoprecipitation analysis revealed that L3MBTL3 and EZH2 interact (*Figure 5C*); and the transient expression of wild-type SET7 in 293T cells promoted the interaction between L3MBTL3 and EZH2, but not an inactive SET7 mutant in which the critical histidine 297 is converted to alanine (H297A, *Figure 5D*). In addition, the SET7-promoted binding of EZH2 to L3MBTL3 is dependent on the presence of K20 in EZH2, as the conversion of K20 to arginine (K20R) in EZH2 (EZH2[K20R]) abolished the EZH2-L3MBTL3 interaction (*Figure 5E*). We also found that while the wild-type EZH2 is reduced by *Kdm1a* silencing, the EZH2[K20R] mutant is resistant to the loss of KDM1A (*Figure 5F*). The S21A mutation of EZH2 (EZH2[S21A]) that converts S21 to alanine remains sensitive to *Kdm1a* silencing, indicating that K20 is still methylated in the EZH2[S21A] mutant protein (*Figure 5F*). Furthermore, co-expression of SET7, L3MBTL3, and DCAF5 are sufficient to promote EZH2 polyubiquitination (*Figure 5G*), but the EZH2[K20R] mutant under the same conditions failed to be polyubiquitinated (*Figure 5H*). These studies collectively indicate that K20 of EZH2 is methylated by SET7, and that L3MBTL3 and CRL4[DCAF5] recognize and target the K20-methylated EZH2 protein for ubiquitination-dependent proteolysis.

## The methylation of K20 and phosphorylation of S21 are mutually exclusive in EZH2

To further examine whether loss of KDM1A affects the protein stability of wild-type EZH2, K20R, and S21A mutant EZH2 proteins, human rhabdoid tumor G401 cells stably expressing the HA-tagged wild-type EZH2, EZH2[K20R], and EZH2[S21A] were established and these cells were transfected with *Kdm1a* siRNA, followed by treating them with protein synthesis inhibitor, cycloheximide, to block translational

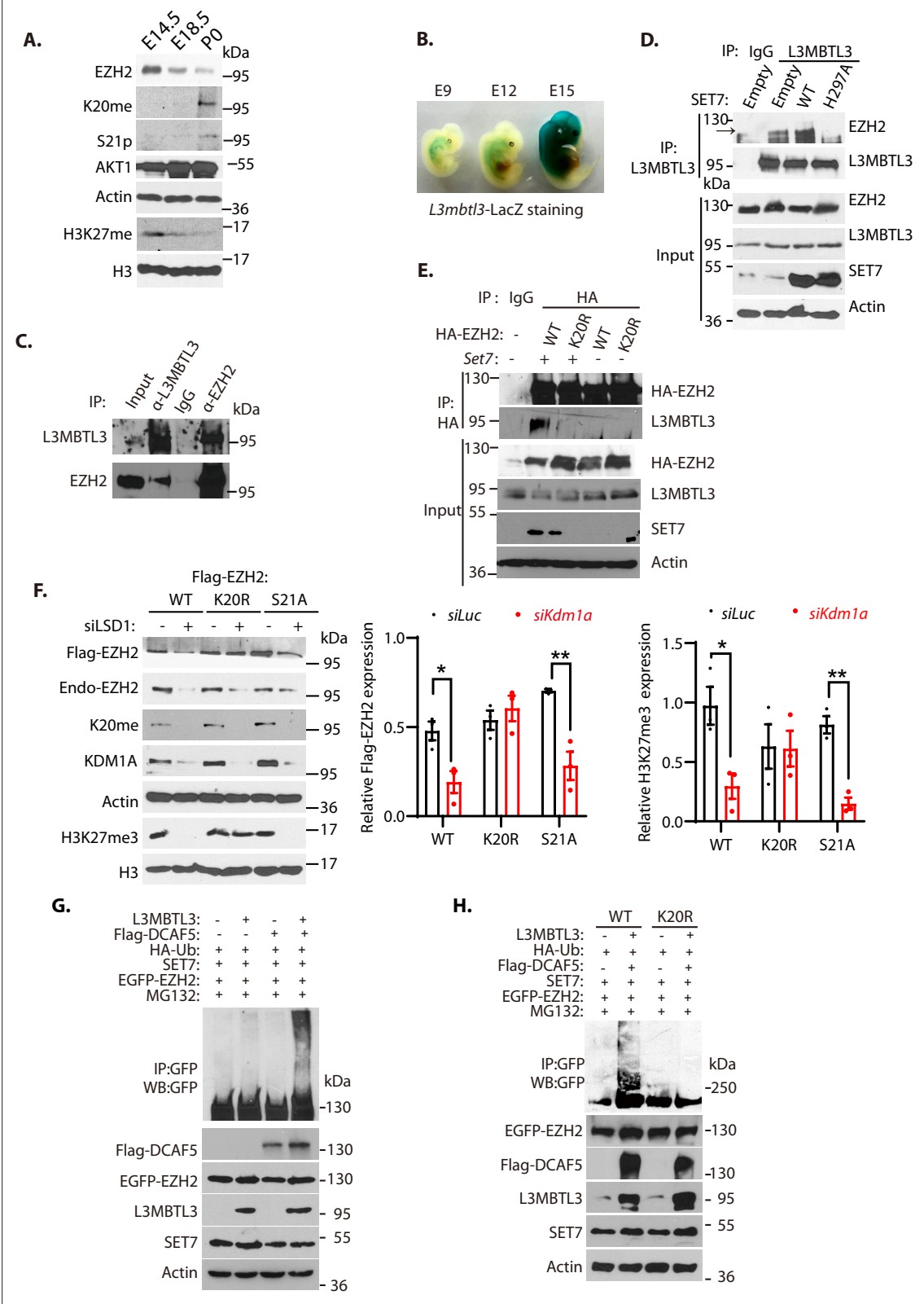

**Figure 5.** L3MBTL3 and DCAF5 target methylated K20 to promote EZH2 proteolysis. (**A**) Mouse embryos at the indicated embryonic days were prepared and the indicated proteins in the lysates were examined with respective antibodies. (**B**) LacZ staining in mouse embryos carrying the LacZ gene under the *L3mbtl* promoter control (*L3mbtl-LacZ*) identifies the endogenous L3MBTL3 expression during mouse development. (**C**) Endogenous L3MBTL3 interacts with EZH2. Lysates were extracted from mouse embryos (E14.5) and the interaction between L3MBTL3 and EZH2 were analyzed

*Figure 5 continued on next page*

*Figure 5 continued*

by co-immunoprecipitation and blotted with respective antibodies. Input: 1/10 of the lysates for Western blotting. (**D**) SET7 stimulates the interaction between L3MBTL3 and EZH2. 293T cells were transiently transfected with the expression vector of *Set7* wild-type, *Set7H297A* mutant, or the empty vector for 48 hr. Cell lysates were immunoprecipitated by anti-L3MBTL3 antibodies and blotted with antibodies against EZH2 and L3MBTL3. (**E**) The K20R mutant does not interact with L3MBTL3. HA-tagged EZH2 or the HA-K20R mutant expressing constructs were co-transfected with vector expressing Set7 or empty vector into 293T cells. Cell lysates were immunoprecipitated with the anti-HA antibody and the blots were immunoblotted with anti-L3MBTL3 and anti-HA antibodies. (**F**) The Flag-tagged *Ezh2* wild-type, *K20r*, or *S21a* mutant were stably expressed in G401 cells. The cells were then transfected with 50 nM siRNAs of luciferase or *Kdm1a* for 48 hr. The protein levels of Flag-EZH2, H3K27me3, and indicated other proteins were analyzed by immunoblotting. The protein bands were quantified and normalized to that of the luciferase control. Significance was indicated as a two-tailed, unpaired, *t*-test. Values are expressed as the mean ± SEM. *p<0.05. **p<0.01. (**G**) and (**H**) The EGFP-tagged wild-type *Ezh2* or *K20r* mutant were co-transfected into 293T cells together with vectors expressing HA-tagged ubiquitin (HA-Ub) and SET7 in the presence or absence of L3MBTL3 and DCAF5 expressing constructs as indicated. Proteasome inhibitor MG132 (5 µg/ml) was added for the last 6 hr to stabilize the polyubiquitinated EZH2. Proteins were immunoprecipitated with anti-GFP antibodies and Western blotted with anti-GFP and other antibodies against indicated proteins.

The online version of this article includes the following source data and figure supplement(s) for figure 5:

**Source data 1.** Original blots for *Figure 5*.

**Source data 2.** Original table sources for quantification of *Figure 5* plots.

**Figure supplement 1.** The mRNA analysis in the mouse embryos.

initiation to measure protein decay rates (***Guo et al., 2022***). We found that the half-lives of EZH2 proteins were reduced after *Kdm1a* silencing in cells expressing the wild-type and S21A mutant EZH2, but the K20R mutant EZH2 was quite resistant to *Kdm1a* silencing (***Figure 6—figure supplement 1***). Since S21 is next to K20 in EZH2 (***Figure 4A and B***), our studies indicated that K20 methylation is affected by S21 phosphorylation (***Figure 4—figure supplement 2D***). However, the initial S21 phosphorylation studies did not reveal that blocking of S21-phosphorylation by AKT inhibition affected the stability of EZH2 protein in T47D cells (***Cha et al., 2005***). Indeed, we found that treatment of T47D cells with MK2206, an AKT inhibitor, inhibited AKT-mediated S21 phosphorylation of EZH2 and increased levels of H3K27me3 due to the reactivation of EZH2 methyltransferase activity as S21-phosphorylation inhibits EZH2 activity, but the total EZH2 protein levels were not significantly altered (***Figure 6A*** and ***Figure 6—figure supplement 2B***), consistent with the original report (***Cha et al., 2005***). To further investigate the regulation of EZH2 protein stability by AKT phosphorylation on S21, we examined the response of primary wild-type MEFs to MK2206 by measuring EZH2 protein levels. We repeatedly found that MK2206 reduced the levels of EZH2 protein, associated with the decreased levels of the phosphorylated AKT, reduced phosphorylated S21 in EZH2, diminished levels of H3K27me3, concurrent with increased levels of the K20 methylated EZH2 protein in MEFs (***Figure 6B***). Our studies in the primary MEFs suggest that in contrast to the response of T47D cells (***Figure 6A***), these is an inverse relationship between the levels of K20-methylation and S21-phosphorylation of EZH2, and that increased levels of K20 methylation after AKT inhibition promote the proteolysis of EZH2 (***Figure 6B***). We next treated human teratocarcinoma PA-1 cells with various doses of MK2206 and found that increasing concentrations of MK2206 also caused the downregulation of EZH2 protein, reduced the levels of phosphorylated AKT and phosphorylated S21, and consequently decreased the levels of H3K27me3 (***Figure 6C***). The downregulation of AKT-mediated processes are associated with the increased levels of K20 methylation in EZH2 (***Figure 6C***), similar to the response of MEFs to MK2206 (***Figure 6B***). Similar inverse relationship between the levels of K20-methylation and S21-phosphorylation of EZH2 were also observed in human lung carcinoma H1299 cells (***Figure 6—figure supplement 2C***). The results in MEFs, PA-1, and H1299 cells all showed that reduction of S21 phosphorylation in MEFs facilitates the methylation of K20 in EZH2 by SET7, leading to EZH2 proteolysis and consequently reducing the trimethylated H3K27 levels. These studies indicate that EZH2 is regulated by a methylation-phosphorylation switch of EZH2 in MEFs, PA-1, and H1299 cells. However, the K20-methylation-dependent degradation of EZH2 protein is not usually detectable after the MK2206 treatment in T47D cells (***Figure 6A***; ***Cha et al., 2005***). We tried to determine the cause of the differential response of EZH2 to AKT inhibition in T47D cells. We found that ectopic and stable expression of HA-tagged L3MBTL3 in T47D cells conferred the EZH2 proteolytic response to MK2206 (***Figure 6D***). Our analysis revealed that inhibition of AKT leads to the reduced level of EZH2 protein and consequent downregulation of H3K27me3 levels in T47D cells that ectopically express L3MBTL3, similar to that of MEFs, PA-1, and H1299 cells (***Figure 6B–D***, and ***Figure 6—figure supplement 2C***). We

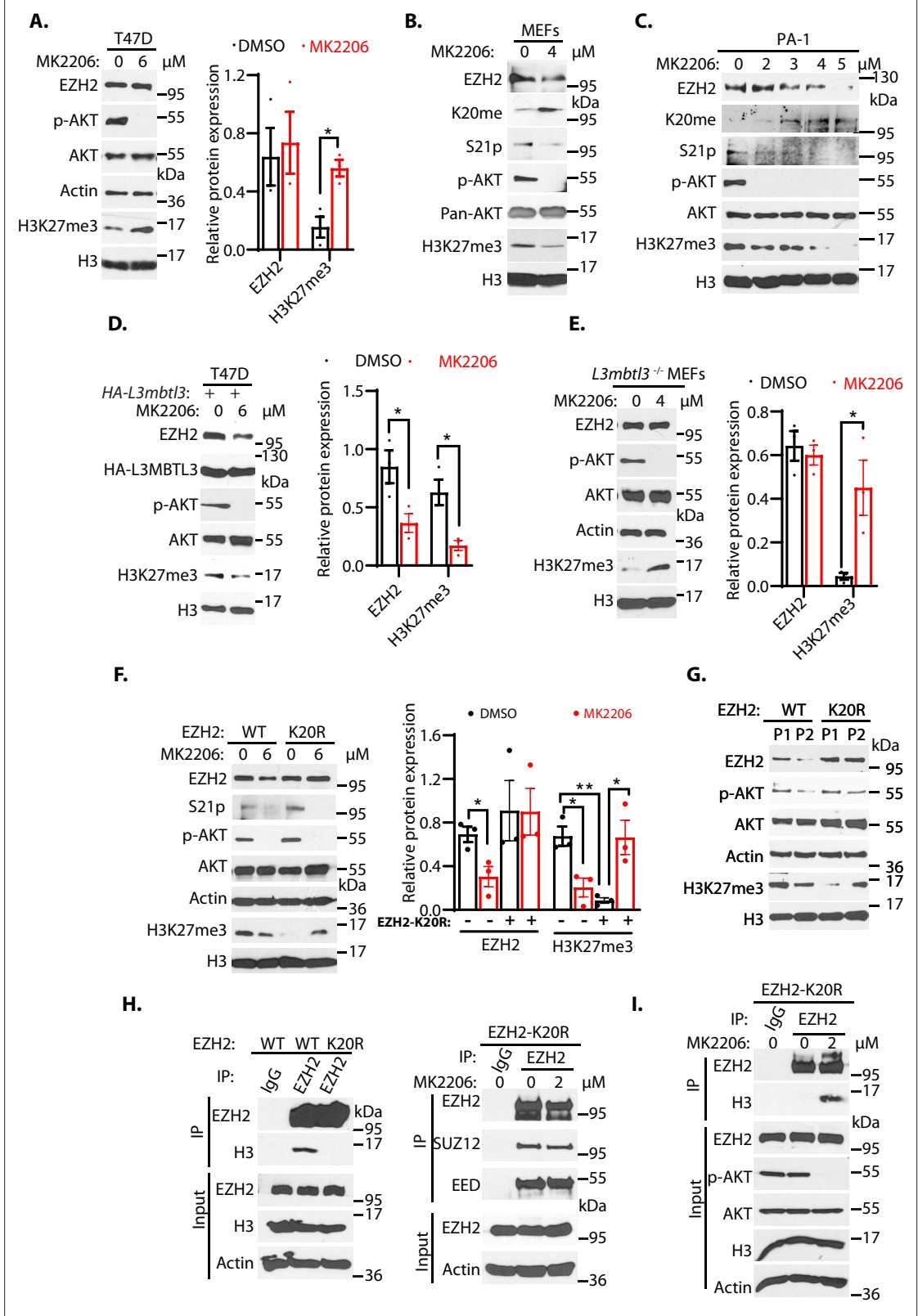

**Figure 6.** The methylation-phosphorylation switch regulates the stability of EZH2. (**A–C**) T47D cells (**A**), mouse embryonic fibroblasts MEFs, (**B**), or PA-1 (**C**) were treated with AKT inhibitor MK2206 (2–6 μM) or dimethyl sulfoxide (control, DMSO) as indicated for 6 hr (**A**) or 4 hr 9 (**B and C**). Protein lysates were blotted with indicated antibodies. (**D**) T47D cells were transfected with the *HA-L3mbtl3* expression construct for 48 hrs and the cells were then treated with or without 6 μM MK2206 for 4 hr. The indicated proteins were blotted with specific antibodies. (**E**) Primary *L3mbtl3* (-/-) MEFs were cultured

*Figure 6 continued on next page*

*Figure 6 continued*

and the cells were treated with or without 4 µM MK2206 for 4 hr. The indicated proteins were blotted with specific antibodies. (**F**) Primary MEFs were obtained from the EZH2 wild-type and homozygous *K20r* mutant embryos (E14.5). They were treated with DMSO or MK2206 for 4 hr. Lysates were prepared and proteins were blotted with indicated antibodies. (**G**) Primary MEFs were obtained from the *Ezh2* wild-type and homozygous *K20r* mutant mouse embryos (E14.5) and cultured as passage 1 (**P1**). They were passaged by splitting 1/3 and cultured as passage 2 (**P2**). Lysates were prepared and proteins were blotted with indicated antibodies. (**H**) Left: lysates from primary *Ezh2* wild-type and *K20r* mutant MEFs were immunoprecipitated with antibodies against EZH2. The proteins were blotted with anti-EZH2 and histone H3 antibodies. Right: primary *Ezh2 K20r* mutant MEF lysates were immunoprecipitated and blotted with anti-EZH2, SUZ12, and EED antibodies as indicated. (**I**) Primary *Ezh2 K20r* mutant MEFs were treated with or without MK2206 (2 µM) for 4 hr. The lysates were immunoprecipitated by anti-EZH2 antibodies. and the blot was blotted with anti-histone H3 antibodies. The input lysates were also blotted for indicated proteins. For (**A**), (**D**), (**E**), and (**F**), protein band intensities were quantified and normalized to that of histone H3 or actin control. Significance was indicated as a two-tailed, unpaired, *t*-test. Values are expressed as the mean ± SEM. *p<0.05. **p<0.01.

The online version of this article includes the following source data and figure supplement(s) for figure 6:

**Source data 1.** Original blots for *Figure 6*.

**Source data 2.** Original table sources for quantification of *Figure 6* plots.

**Figure supplement 1.** Regulation of EZH2 protein stability by K20 methylation with protein decay assays.

**Figure supplement 1—source data 1.** Original blots for *Figure 6—figure supplement 1*.

**Figure supplement 2.** L3MBTL3 levels in various cancer cell lines.

**Figure supplement 2—source data 1.** Original blots for *Figure 6—figure supplement 2*.

**Figure supplement 3.** Schematic demonstration of the mouse *Ezh2 K20r* knock-in mutagenesis and a representative DNA sequencing profile performed on the mouse *K20r* mutant of EZH2.

**Figure supplement 4.** A fraction of K20R mutant protein is in the cytoplasm.

**Figure supplement 4—source data 1.** Original photos and blots for *Figure 6—figure supplement 4*.

reasoned that EZH2 protein levels in T47D cells did not significantly respond to AKT inhibition is likely caused by the weak L3MBTL3-dependent proteolysis activity of EZH2 in this cell line, and we tried to further investigate this possibility by using the *L3mbtl3* null MEFs that are deficient in targeting the K20-methylation-dependent proteolysis of EZH2 (*Figure 2B*). Our studies revealed that MK2206 treatment caused increased levels of H3K27me3 in the *L3mbtl3* null MEFs due to the removal of S21 phosphorylation-mediated inhibition on EZH2 and PRC2, whereas AKT inhibition did not cause any further changes in EZH2 protein levels because of *L3mbtl3* deletion, a result similar to the response of T47D cells to MK2206 (*Figure 6A and E*). These studies are consistent with our hypothesis that T47D cells may have reduced activities in L3MBTL3 and CRL4$^{DCAF5}$ ubiquitin E3 ligase activities to target the K20-methylated EZH2 for proteolysis. Our results are also consistent with other reports showing that AKT inhibition by MK2206 reduced EZH2 protein stability in several cancer cells (*Riquelme et al., 2016*), indicating the methylation-phosphorylation switch of EZH2 exists in many cells, but T47D is defective in this pathway. *L3mbtl3* is mutated in medulloblastoma and is further implicated in other pathological disorders such as multiple sclerosis, insulin resistance, prostate cancer, and breast cancer (*Andlauer et al., 2016*; *Bonasio et al., 2010*; *Kar et al., 2016*; *Lotta et al., 2017*; *Northcott et al., 2009*). Our studies found that the levels of L3MBTL3 protein are differentially expressed in various cancer cell lines and that T47D cells express relatively low levels of L3MBTL3 protein among the cell lines we analyzed (*Figure 6—figure supplement 2A*). Since alterations of EZH2 and PRC2, including EZH2 overexpression (*Chase and Cross, 2011*; *Völkel et al., 2015*; *Zeng et al., 2022*), are frequently associated with a wide variety of human cancers, it is likely that the lysine methylation-dependent proteolysis in cancer cells is affected at various levels, including the altered L3MBTL3 protein levels, to prevent EZH2 proteolysis. Further investigation is required to clarify the alterations of L3MBTL3 and substrate sensitivity in various cancer cells.

## Establishment and characterization of *Ezh2*^K20R mutant mice

To systematically determine the effect of the K20 methylation on EZH2 and its relationship to S21 phosphorylation, we employed the CRISPR-Cas9 gene editing technique to change the nucleotides AA for codon 20 of the mouse *Ezh2* gene to GG, resulted in converting K20 to arginine (K20R) in the mouse (*Kim et al., 2014a*; *Kleinstiver et al., 2016*; *Slaymaker et al., 2016*). The targeted K20R knock-in allele (*Ezh2*^K20R) was confirmed by direct DNA sequencing of the heterozygous and homozygous *Ezh2*^K20R mouse strains (*Figure 6—figure supplement 3*). The primary homozygous *Ezh2*^K20R

MEFs and wild-type control MEFs were isolated and cultured from the homozygous Ezh2[K20R] and wild-type mouse embryos. These MEFs were treated with MK2206 and the wild-type and K20R of EZH2 proteins were examined. Our results revealed that while the EZH2 and H3K27me3 proteins in wild-type MEFs were reduced by MK2206, the K20R mutant of EZH2 was resistant to AKT inhibition (Figure 6F). However, the control-treated primary Ezh2[K20R] MEFs showed lower H3K27me3 levels than that of the wild-type control MEFs, likely because the S21 residue of the EZH2[K20R] protein can still be phosphorylated by the AKT activity to inhibit PRC2 catalytic activity for the H3K27me3 levels. Consistent with this possibility, MK2206 treatment induced the increased levels of H3K27me3 in the Ezh2[K20R] MEFs (Figure 6F), similar to the response of T47D cells and the L3mbtl3 null MEFs to MK2206 (Figure 6A and E). Previous studies reported that the protein levels of EZH2 decline during the in vitro passages of primary MEFs (Bracken et al., 2007). We tried to determine whether the EZH2 protein level decline is dependent on K20 methylation by culturing both Ezh2[K20R] and the corresponding wild-type MEFs. Our studies indicate that while the in vitro cell passage from passage 1 to passage 2 induces the down-regulation of wild-type EZH2 protein and H3K27me3 levels in the primary MEFs, the EZH2[K20R] protein is resistant to the passage-dependent decline (Figure 6G). In addition, the H3K27me3 levels increased in the passage 2 of the Ezh2[K20R] MEFs from the lower levels in the passage 1 of the Ezh2[K20R] MEFs, likely caused by the downregulation of phosphorylated and active AKT in the passage 2 of the Ezh2[K20R] MEFs (Figure 6G). We tried to examine the changes of S21 phosphorylation during the cell passages but the S21 phosphorylation signal is too low to be detected by our anti-S21 phosphorylation antibodies. Nevertheless, our results indicate that the K20R mutation stabilizes EZH2 protein to increase H3K27 trimethylation by PRC2. Loss of AKT phosphorylation on S21 of EZH2 promotes K20 methylation and EZH2 proteolysis, whereas loss of K20 methylation conversely facilitates S21 phosphorylation and stabilizes EZH2 protein.

Since S21 phosphorylation was reported to reduce EZH2 interaction with histone H3 (Cha et al., 2005), we analyzed the interaction between EZH2 and histone H3 by immuno-co-precipitation. Our results showed that the EZH2[K20R] protein reduced its association with histone H3 than the wild-type EZH2 protein in MEFs, although their interaction with SUZ12 or EED did not change (Figure 6H). It is likely that the K20R mutation in EZH2 may facilitate the phosphorylation of S21 by AKT to block the interaction between EZH2 and histone H3. Indeed, we found that AKT inhibition enhanced the interaction between the EZH2[K20R] mutant and histone H3 (Figure 6I). It has been shown that the Notch1 intracellular domain (NICD) increases the cytoplasmic EZH2 levels during early megakaryopoiesis due to AKT-dependent phosphorylation of EZH2 (Roy et al., 2012). We examined the distribution of the EZH2[K20R] mutant in MEFs. Immunofluorescent staining revealed that while the wild-type EZH2 protein was nuclear with little cytoplasmic presence, a small fraction of the EZH2[K20R] protein was found to localize to the cytoplasm (Figure 6—figure supplement 4A), although the majority of EZH2[K20R] still remained in the nucleus. We also performed cell fractionation studies, and our biochemical analysis indicated that a fraction of the EZH2[K20R] protein was localized in the cytoplasmic fraction, but the majority of EZH2[K20R] protein was still in the nucleus (Figure 6—figure supplement 4B). We also found that in Ezh2[K20R] MEFs, the elevated levels of EZH2 protein are also accompanied by the increased levels of SUZ12 protein (Figure 6—figure supplement 4C). Thus, our results indicate that the K20R mutation abolishes the methylation of K20 in EZH2 and stabilizes EZH2 protein, and consequently promotes S21 phosphorylation by AKT to prevent EZH2 to interact with histone H3, and that the K20R mutation promotes the presence of a small fraction of EZH2 in the cytoplasm, likely due to the increased S21-phosphorylation in the EZH2[K20R] mutant. Further investigation is required to determine the function and regulation of cytoplasmic EZH2[K20R] protein.

## The K20R mutation causes the expansion of bone marrow hematopoietic populations

To determine the in vivo functional deficit of the K20R mutation of EZH2, we bred the heterozygous Ezh2[K20R/+] mice to generate the homozygous K20R knock-in mutant (Ezh2[K20R/K20R]) mice. The homozygous Ezh2[K20R/K20R] progeny mice are viable with normal Mendelian genetic ratios. We found that the livers and spleens of the 8-month-old homozygous Ezh2[K20R/K20R] mice are enlarged (hepatosplenomegaly), as compared with those of the wild-type littermates (Figure 7A and B). In addition, the total cell numbers of the bone marrow (BM) in 5-month-old mice were significantly increased in the Ezh2[K20R/K20R] mice than in that of wild-type animals (Figure 7C). We next investigated the potential

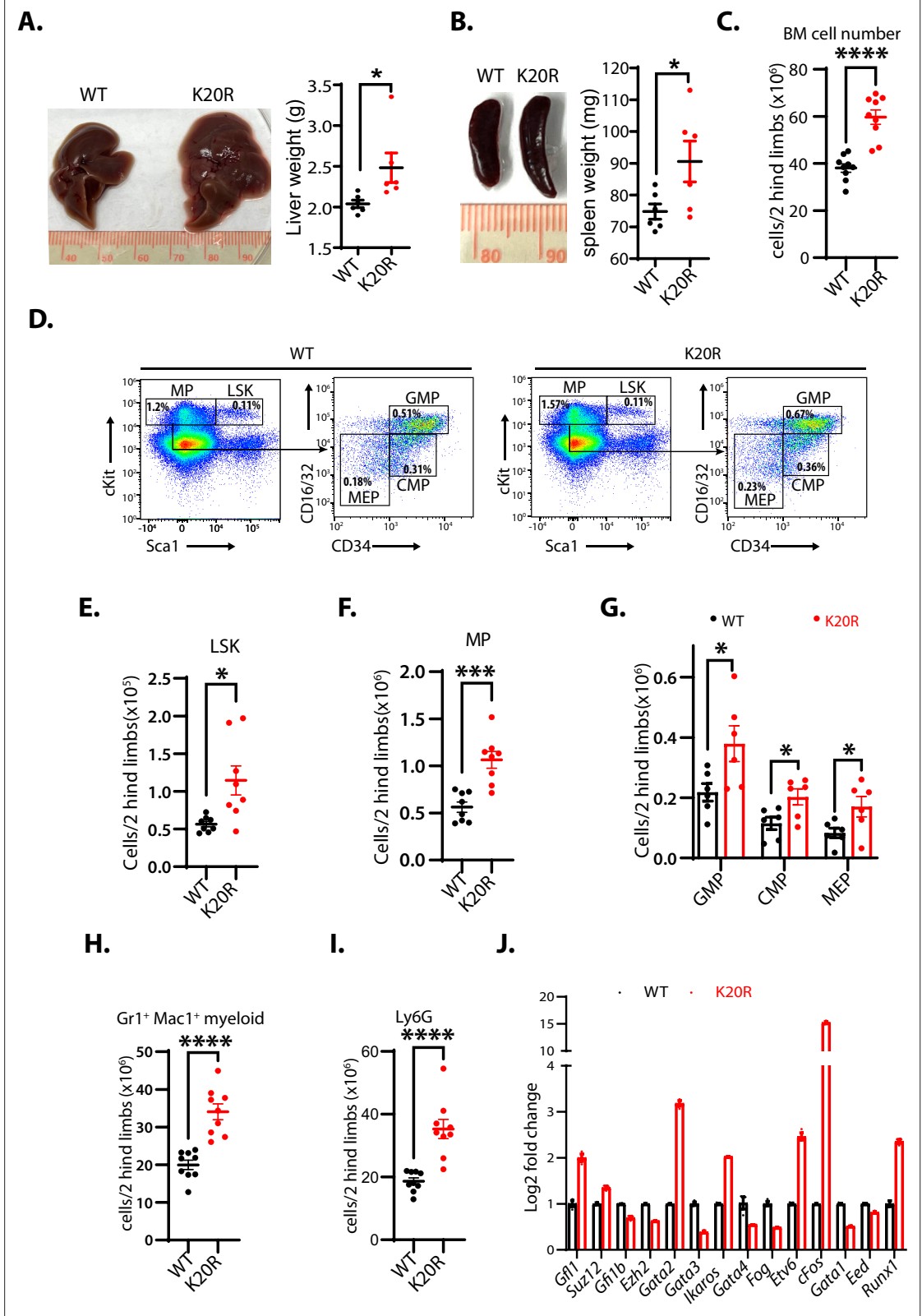

**Figure 7.** K20R mutation causes hepatosplenomegaly and expansion of hematopoietic populations. (**A and B**) The enlarged liver or spleen from the homozygous *K20r* mutant mice, as compared to that of the *Ezh2* wild-type mice (8 months old). The weights of livers or spleens from the *K20r* and wild-type *Ezh2* mice (7~10-month-old) were measured and plotted. Values are means ± SEM (N=6, *P<0.05). (**C**) Number of total cells in bone marrows harvested from 5-month-old *K20r* and wild-type *Ezh2* mice. Values are means ± SEM (n=9, ****p<0.0001). (**D–F**) Representative flow cytometric

*Figure 7 continued on next page*

*Figure 7 continued*

(FACS) profiles of bone marrow cells from the *K20r* and *Ezh2* wild-type mice. Flow cytometry plots were gated on the Lin⁻cKit⁺ (myeloid progenitors) subpopulation that is subclassified into common myeloid progenitor (CMP), granulocyte-monocyte progenitor (GMP), and myeloid erythroid progenitor (MEP) based on Lin⁻cKit⁺ CD16/32 and CD34 expression. The number of immature cells Lin⁻cKit⁺Scal1⁺(**E**), Lin⁻cKit⁺ (**F**), CMP, GMP, and MEP in (**G**), and differentiated cells Mac1⁺Gr1⁺ myeloid (**H**) and Ly6.6G myeloid (**I**) in bone marrow samples harvested from the *Ezh2* wild-type and *K20r* mutant mice. Values are means ± SEM (n=6–9). Significance was indicated as a two-tailed, unpaired, *t-test*. *$p<0.05$.***$p<0.001$. ****$p<0.0001$. (**J**) The quantitative RT-PCR (qRT-PCR) analysis shows mRNA expression levels of indicated genes in bone marrow samples harvested from the *Ezh2* wild-type and *K20r* mutant mice.

The online version of this article includes the following source data and figure supplement(s) for figure 7:

**Source data 1.** Original photos for *Figure 7*.

**Source data 2.** Original table sources for quantification of *Figure 7* plots.

**Figure supplement 1.** Effects of K20R mutation in T and B cells.

**Figure supplement 2.** Other defects of *K20r* mutation in the mouse.

**Figure supplement 2—source data 1.** Original photo for *Figure 7—figure supplement 2*.

role of the K20R mutation of EZH2 in regulating the primitive hematopoietic subpopulations in the bone marrow. The homozygous *Ezh2*^K20R/K20R mice contained a significantly higher absolute number of the Lin⁻Sca1⁺c-Kit⁺ (LSK) cells and LK cells, as compared with that of the wild-type control mice (*Figure 7D–F*). The LSK cells represent multipotent stem cells that are CD48⁺, CD71⁺, and enriched for CD150⁺ capable of regenerating bone marrow from irradiated mice, whereas the LK cells are c-Kit^low and enriched for Sca-1⁺ progenitor cells (*Adolfsson et al., 2001*; *Béguelin et al., 2013*). Analysis of the LK cells of the *Ezh2*^K20R/K20R mice revealed an obvious expansion in the numbers of granulocyte-monocyte progenitor (GMP), common myeloid progenitor (CMP), and myeloid erythroid progenitor (MEP) populations (*Figure 7G*). Moreover, our flow cytometric analysis revealed that the bone marrow of the *Ezh2*^K20R/K20R mutant mice showed an increased proportion of mature myeloid cells (Gr1⁺/Mac1⁺ and Ly6G) (*Figure 7H and I*), whereas the absolute numbers of B and T lymphocytes remained similar to that of the wild-type animals (*Figure 7—figure supplement 1A–C*). Our quantitative reverse transcription-polymerase chain reaction (RT-PCR) analysis further showed that the homozygous *Ezh2*^K20R/K20R mutant bone marrow cells displayed significantly increased RNA expression levels of transcriptional regulators for the hematopoietic stem cell (HSC) activities, specification, and self-renewal, including *Runx1*, *Gata2*, *Etv6*, *cFos*, *Ikaros*, and *Gfi1*, whereas our RT-PCR examination of previously reported EZH2-repressed genes revealed that some of them are downregulated, such as *Strc*, *Syngap1*, *Bmi1*, *Ltgb5*, *Ppfia4*, and *Runx3* (*Figure 7J* and *Figure 7—figure supplement 1D*; *Deneault et al., 2009*; *Hock et al., 2004*; *North et al., 1999*; *Pereira et al., 2013*; *Ross et al., 2012*; *Tsai et al., 1994*; *Tu et al., 2021*). In addition to the altered hematopoietic system in *Ezh2*^K20R/K20R mutant mice, we found that the xiphoid process of *Ezh2*^K20R/K20R mice is more pronounced and harder than that of the wild-type mice (*Figure 7—figure supplement 2*). Further investigation is required to analyze the effect of EZH2^K20R mutant on the xiphoid alteration.

## Phenotypic characterization of the *Ezh2*^K20R/K20R mutant mice

Many studies have shown that EZH2 overexpression is correlated with human hematological malignancies, and EZH2 has been established as a critical target for hematological cancers (*Abd Al Kader et al., 2013*; *van Galen et al., 2007*; *Yan et al., 2013*). Since the *Ezh2*^K20R/K20R mutant mice exhibit enlarged spleen and liver (*Figure 7A and B*), our hematoxylin and eosin (HE) staining of the splenic sections from 8-month-old *Ezh2*^K20R/K20R mutant mice revealed the prominent hemosiderin-laden macrophages in the red pulp and mild to moderate reactive hyperplasia in the follicles of the white pulp (*Figure 8A*). The *Ezh2*^K20R/K20R mutation also caused mild to moderate diffuse cytoplasmic vacuolation in the mouse liver sections (*Figure 8A*). Furthermore, increased protein levels of EZH2 and H3K27me3 were observed in the spleen of adult *Ezh2*^K20R/K20R mice (*Figure 8C*), associated with the increased Ki67 immunostaining of splenic sections, indicating increased numbers of proliferating cells in *Ezh2*^K20R/K20R mutant animals (*Figure 8B*). These findings suggest that the single amino acid substitution of lysine-to-arginine mutation, K20R, in EZH2 modulates EZH2 protein stability and induces mild-to-moderate reactive hyperplasia in animals. To test whether the phenotypes of the *Ezh2*^K20R/K20R mutation are associated with altered transcriptional factors in the spleen, we performed quantitative RT-PCR analysis of

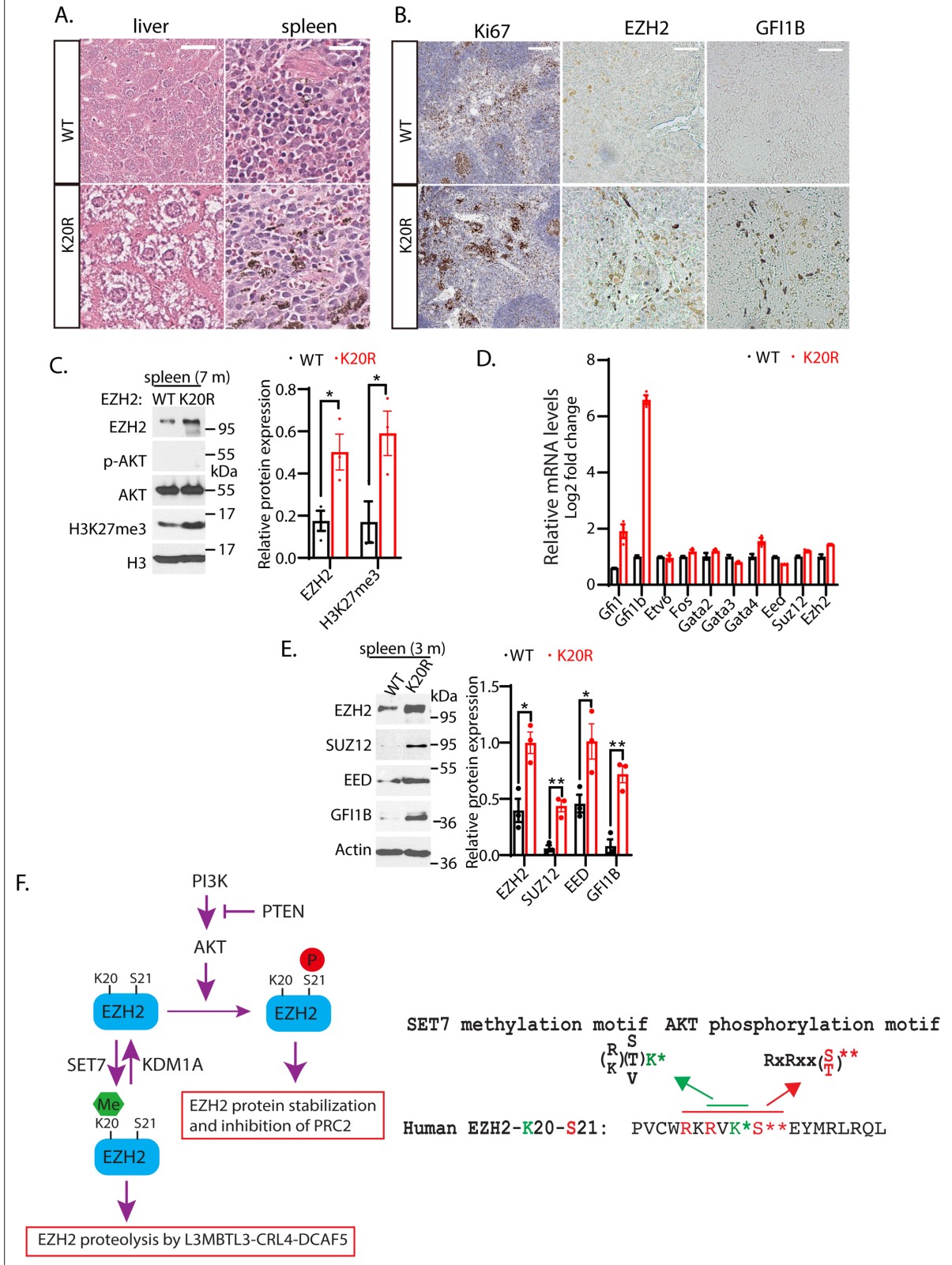

**Figure 8.** K20R mutation induces hyperplasia in the mouse spleen. (**A**) Hematoxylin and eosin (H&E) histological staining of liver and spleen sections from the *K20r* and *Ezh2* wild-type mice (8-month-old). Scale bar: 200 µm. (**B**) The anti-Ki67 and anti-GFI1B immunostaining of spleen sections from the *K20r* and *Ezh2* wild-type mice (8-month-old). Scale bar: 50 mm. (**C**) Left: protein lysates were extracted from the spleen of the *K20r* and *Ezh2* wild-type mice and blotted with indicated antibodies. Right: band intensities were quantified and normalized to that of histone H3 control. Significance was

*Figure 8 continued on next page*

*Figure 8 continued*

indicated as a two-tailed, unpaired, *t*-test. Values are expressed as the mean ± SEM. *p<0.05. (**D**) RNA levels of indicated hematopoietic regulatory genes were measured by quantitative RT-PCR (RT-qPCR) from the spleens of the *Ezh2* wild-type and *K20r* mice (4-month-old). The mRNA levels were measured in triplicate by RT-qPCR. (**E**) Protein lysates were extracted from the spleens of the *Ezh2* wild-type and *K20r* mice and detected with anti-EZH2, SUZ12, EED, and GFI1B antibodies, using the actin antibody as a control. The protein band intensity values are means ± SEM (n=3). Significance is indicated as a two-tailed, unpaired, *t*-test. *p<0.05.**p<0.01. (**F**) Model: Left panel: The lysine residue 20 (**K20**) of EZH2 is methylated by SET7 methyltransferase and the level of methylated K20 is reversibly removed by KDM1A demethylase. L3MBTL3 preferentially binds to the methylated K20 in EZH2 to recruit the CRL4[DCAF5] ubiquitin E3 ligase complex to target the methylated EZH2 for ubiquitin-dependent proteolysis. Right panel: The K20 methylation is negatively regulated by the phosphorylation of serine 21 (**S21**) by the PI3K-activated AKT. Conversely, the S21 phosphorylation is mutually exclusive to the methylation of K20, resulting in the methylation-phosphorylation switch to control the activity and proteolysis of EZH2 for H3K27 trimethylation.

The online version of this article includes the following source data and figure supplement(s) for figure 8:

**Source data 1.** Original photos and bots for *Figure 8*.

**Source data 2.** Original table sources for quantification of *Figure 8* plots.

**Figure supplement 1.** The list of DNA oligonucleotide primers for RT-PCR.

---

the spleens from the wild-type and *Ezh2*[K20R/K20R] mice (*Figure 8D*). Different from the expanded bone marrow cells in the *Ezh2*[K20R/K20R] mutant, the quantitative RT-PCR analysis of the *Ezh2*[K20R/K20R] spleen cells showed significantly increased expression of *Gfi1b* (*Figure 8D*), an essential proto-oncogenic transcriptional regulator necessary for development and differentiation of erythroid and megakaryocytic lineages (*Pereira et al., 2013*; *Ross et al., 2012*). We also assessed and compared the protein levels of GFI1B in the *Ezh2*[K20R/K20R] and the wild-type mouse spleens. Increased GFI1B protein levels were detected in the *Ezh2*[K20R/K20R] mutant spleens by immunohistological staining with anti-GFI1B antibodies (*Figure 8B*). The increased GFI1B protein in the *Ezh2*[K20R/K20R] mutant spleens were also confirmed by Western blotting analyses of 3- or 7-month-old *Ezh2*[K20R/K20R] mice (*Figure 8E*). GFI1B is a transcription repressor that has GFI1/GFI1B binding sites close to its mRNA start site and can repress its own transcription (*Vassen et al., 2005*). Previous studies have shown that GFI1B, GATA1, and EZH2 physically interact and cooperate to suppress target genes such as *Hes1* promoter (*Pinello et al., 2014*; *Ross et al., 2012*; *Yu et al., 2009*). It remains to be further characterized how *Gfi1b* expression is induced by K20R mutation of EZH2. Our studies are consistent with a model by which K20 of EZH2 protein is methylated by SET7 to recruit L3MBTL3 and the CRL4[DCAF5] ubiquitin ligase complex to target the EZH2 protein for ubiquitin-dependent degradation, and that KDM1A serves as a demethylase to remove the methyl group from the methylated EZH2 to prevent EZH2 degradation to preserve the integrity of the PRC2 complex (*Figure 8F*). Importantly, the K20 methylation-dependent proteolysis of EZH2 is negatively regulated by the AKT-mediated phosphorylation of S21, which acts to inhibit the H3K27 methyltransferase activity of PRC2 (*Figure 8F*).

## Discussion

The alterations of EZH2 and PRC2 are frequently associated with a wide variety of human cancers, including hematopoietic malignancies, and the EZH2 inhibitor, tazemetostat, has been approved for the treatment of follicular lymphoma (*Duan et al., 2020*). However, how the levels of EZH2 and PRC2 are regulated remains largely unclear. We have previously shown that EED in PRC2 interacts with CRL4, but the significance of this interaction remains unclear (*Higa et al., 2006*). In this report, we found that the protein stability of EZH2 is dynamically regulated by a novel lysine methylation-dependent proteolysis involving the activities of SET7, KDM1A, L3MBTL3, and the CRL4[DCAF5] ubiquitin ligase complex. Our studies revealed that K20 of EZH2 is monomethylated by SET7 methyltransferase, and the methylated K20 serves as a substrate of KDM1A demethylase. The methylated K20 is recognized by specific methyl lysine reader L3MBTL3 to promote EZH2 for ubiquitin-dependent proteolysis by the CRL4[DCAF5] ubiquitin E3 ligase complex, resulting in the disassembly of the PRC2 complex and reduction of H3K27 trimethylation in animals, MEFs, and cancer cells. Since K20 methylation of EZH2 destabilizes the histone methyltransferase while S21 phosphorylation impairs EZH2 enzymatic activity, it is likely that the fraction of EZH2 that is not modified by K20 methylation or S21 phosphorylation is the active form for H3K27 trimethylation. On the other hand, the K20-methylated fraction of EZH2 is protected by PHF20L1, as our studies showed that the loss of PHF20L1 destabilizes EZH2 protein.

The PHF20L1-protected, K20-methylated EZH2 fraction may also be catalytically active. Interestingly, the methylation of K20 is prevented by the AKT-dependent phosphorylation of S21 in EZH2. As the active PRC2 complex contains the unphosphorylated EZH2 at S21, which is methylated at K20 to be targeted for proteolysis, further investigation is required to determine how the methylation/phosphorylation switch operates during normal development and how this regulation is altered in various human diseases including cancers.

## Materials and methods

Cells: Human lung carcinoma H1299 (NCI-H1299, CRL-5803) were obtained from American Type Culture Collection (ATCC) and authenticated by lack of p53; cervical carcinoma HeLa (CRM-CCL-2), obtained from ATCC and authenticated by high levels of CDK inhibitor CDKN2A and p53; embryonic kidney 293T (CRL-3216), obtained from ATCC and authenticated by high levels of CDK inhibitor CDKN2A and p53; colon cancer HCT116 (CCL-247), obtained from ATCC and authenticated by expression of wild-type p53 and induction of CDKN1A by UV irradiation; rhabdoid tumor G401 (CRL-1441), obtained from ATCC and authenticated by lack of expression of SMARCB1; breast cancer T47D (HTB-133), obtained from ATCC and authenticated by lack of ARID1A expression; teratocarcinoma PA-1(CRL-1572), obtained from ATCC and authenticated by expression of SOX2 and OCT4; squamous cell lung carcinoma H520 (NCI-H520, HTB-182), obtained from ATCC and authenticated by high expression of SOX2, were cultured in RPMI-1640, McCoy's 5 a, Eagle's Minimum Essential Medium, or DMEM medium with 10% FBS and 1% antibiotics as described (*Guo et al., 2022*; *Leng et al., 2018*; *Zhang et al., 2013*). Mouse embryonic fibroblasts (MEFs) were generated from wild-type and *L3mbtl3* deletion mutant mouse embryos, or the CAGGCre-ER/*Kdm1a*fl/fl and CAGGCre-ER/*Kdm1a*fl/fl/*L3mbtl3*fl/fl, wild-type and *Ezh2*K20R/K20R knock-in mouse embryos (E12.5-E13.5), as according to approved IACUC protocols (IACUC-01161)711621 and (IACUC-01177)832146 described previously (*Guo et al., 2022*). All cell lines or MEFs are tested for mycoplasma contamination. For stable expression, human *Ezh2* wild-type and the K20R or S21A mutant of human *Ezh2* were cloned into the retroviral pMSCV-Puro vector containing 3xFlag-3xHA epitope (Addgene) and the recombinant retroviruses were packaged in 293T cells (*Guo et al., 2022*; *Leng et al., 2018*). Viral-infected H1299 and G401 cells were selected by puromycin resistance as described before (*Guo et al., 2022*; *Leng et al., 2018*).

### Peptide synthesis and preparation of methylated peptides

The methylated K17-(KSEKGPVCWRK(me1)RVKSEYMRLRQLKRFRRAD), K20-(KSEKGPVCWRKRVK(me1)SEYMRLRQLKRFRRAD), K20-(KSEKGPVCWRKRV-K(me2)SEYMRL-RQLKRFRRAD), K20- (KSEKGPVCWRKRVK(me3)SEYMRLRQLKR-FRRAD), and cognate unmethylated peptides of EZH2 were synthesized at ABI Scientific (*Guo et al., 2022*). The monomethylated K20 peptide was used to raise rabbit polyclonal antibodies after coupling the peptide to keyhole limpet hemocyanin (KLH)(*Guo et al., 2022*). Affinity purification of methylated peptide antibodies were conducted as described (*Guo et al., 2022*; *Leng et al., 2018*). The unmethylated and monomethylated K20 peptides were immobilized to Sulfolink-coupled-resins (Thermo Fisher) by covalently cross-linking with the cysteine residues at the end of the peptides to the resin (*Leng et al., 2018*). The anti-monomethylated K20 peptide sera (5 ml each) were diluted in 1:1 in PBS and first passed through the unmethylated K20 peptide columns (1 ml) for three times to deplete anti-K20 peptide antibodies (*Guo et al., 2022*). The unbound flow-through antibody fractions were then loaded onto the monomethylated K20 peptide column (0.5 ml), washed, and the bound antibody fractions were eluted by 5 ml of 100 mM glycine, pH2.5. The eluted antibodies (0.5 ml/fraction) were immediately neutralized by adding 100 µl of 2 M Tris, pH8.5, and tested for specificity towards the monomethylated K20 peptide but not to the unmethylated K420 peptide (*Guo et al., 2022*; *Leng et al., 2018*; *Zhang et al., 2019*). Human *Kdm1a* were cloned into pGEXKG vectors and purified by GSH-Sepharose (GE Healthcare). For demethylation reaction, purified 1 µg of control GST control or GST-KDM1A proteins were incubated with 100 ng of the unmethylated or monomethylated K20 peptides for 4 hr at room temperature, and the resulting peptides were blotted onto nitrocellulose membrane (*Guo et al., 2022*; *Leng et al., 2018*). The demethylated peptides were detected by immuno-blotting with affinity-purified anti-monomethylated K20 antibodies.

## Antibodies and immunological analysis

Anti-KDM1A (A300-215A), anti-L3MBTL3 (A302-852), anti-SUZ12 (A302-407A), and anti-SET7 (A301-747A) antibodies were purchased from Fortis Life Sciences. Anti-EED (ab236292) was purchased from Abcam. Phospho-EZH2 (Ser21) antibody (AF3822) was purchased from Affinity Biosciences. Anti-EZH2 (5246), anti-EED (85322), anti-GFI1B (5849), and anti-H3K27me3 (9733) were from Cell Signaling Technology. Actin (Sc-1616) antibody was purchased from Santa Cruz Biotechnologies. Anti-Flag, ant-HA, and anti-GFP antibodies were purchased from Sigma. Anti-GAPDH (60004–1-Ig) antibody was purchased from Proteintech. Rabbit anti-L3MBTL3 and affinity-purified anti-DCAF5 antibodies were also produced in the laboratory as previously described (*Leng et al., 2018*). For direct Western blotting, cells were lysed in the 1 X SDS sample buffer (4% SDS, 100 mM Tris, pH6.8, and 20% glycerol), quantified by protein assay dye (Bio-Rad), and equalized by total proteins (*Leng et al., 2018*). For immunoprecipitation (IP), cells were lysed with an NP40-containing lysis buffer (0.5% NP40, 50 mM Tris, pH 7.5, 150 mM NaCl, and protease inhibitor cocktails) (*Leng et al., 2018*). About 500 μg of lysates and 1 μg antibody were used for each IP assay. The antigen–antibody complexes were pulled down by 30 μl Protein A-Sepharose (GE Healthcare) and specific proteins were detected by the Western blotting analysis, using secondary goat anti-mouse HRP (Jackson Immuno Research, 115-035-008) and goat–anti-rabbit antibodies (Jackson Immuno Research, 111-035-008), or Protein A HRP (GE Healthcare, NA9120V), all at 1:2500 dilutions (*Leng et al., 2018*; *Zhang et al., 2019*).

## Transfection and siRNAs

Oligofectamine was used for siRNA silencing in HeLa, H1299, HCT116, G401, or 293T cells, whereas Lipofectamine 2000 was used for transient transfection as described previously (*Leng et al., 2018*; *Zhang et al., 2019*; *Zhang et al., 2013*). Typically, 50 nM of each siRNA or their combinations were transfected into target cells for 48 hr and cells were directly lysed in the 1 X SDS lysis sample buffer (*Leng et al., 2018*). For verification of the silencing effects of various target proteins, usually two or three independent siRNAs were designed to examine the knockdown efficiency and the consequences of knockdown on target proteins (*Guo et al., 2022*; *Leng et al., 2018*; *Zhang et al., 2019*). The siRNAs for human genes are: *Kdm1a*: GGAAGAAGAUAGUGAAAAC; *Kdm1a*-2: UGAAAACUCAGGAAGAUU; *Kdm1a*-3'UTR: GGGAGGAACUUGUCCAUUA; *Dcaf5-1*: CUGCAGAAACCUCUACAA; *Dcaf5-2*: ATCACCAACTTCTGACATA; *L3mbtl3-1*: GATGCAGATTCTCCTGATA; *L3mbtl3-2*: GGTA CCAACTGCTCAAGAA; *Set7-1*: GGGCAGTATAAAGATAACA; *Set7-2* SMART pool: CAACUGCAUCUACGAUAU, CCUGGACGAUGACGGAUUA, GGAGUGUGCUGGAUAUAUU, and CAAACUGGCUACCCUUAUG (*Guo et al., 2022*; *Leng et al., 2018*; *Zhang et al., 2019*). All siRNAs were synthesized from Horizon Discovery.

## *L3mbtl3* deletion in HCT116 cells

The homozygous deletion of human *L3mbtl3* alleles were conducted using the CRISPR-Cas9 gene edition with the lentiviral plasmid lentiCRISPRv2 (AddGene 52961) and the human *L3mbtl3* gRNA, GATTCGGCTGTACTAAAGCA, and the packaging plasmids pVSVg (AddGene 8454), and psPAX2 (AddGene 12260) transfected into 293T cells (*Doench et al., 2014*; *Sanjana et al., 2014*; *Shalem et al., 2014*). The gRNA sequences were designed by using http://www.broadinstitute.org/rnai/public/analysis-tools/sgrna-design (*Doench et al., 2014*) and the GeCKO Lentiviral CRISPR Tool Box (https://media.addgene.org/cms/filer_public/4f/ab/4fabc269-56e2-4ba5-92bd-09dc89c1e862/zhang_lenti-crisprv2_and_lentiguide_oligo_cloning_protocol_1.pdf). Other single *L3mbtl3* deletion clones with anti-sense gRNAs, GTAGCAACACAGATGAATGA or GTACCTGTGGGACATCCAGG, were also similarly obtained. The packaged recombinant lentivirus particles were used to infect HCT116 cells and selected for puromycin-resistant colonies. The single-cell clones with homozygous deletion of human *L3mbtl3* alleles were identified by DNA sequencing. The effects on EZH2 were similar in all these *L3mbtl3* deletion clones.

## Animals

The *Kdm1a*^{fl/+} conditional mutant (B6.129-*Kdm1a* tm1.1Sho/J; Strain #: 023969) (*Kerenyi et al., 2013*), transgenic actin-Cre-ER (CAGGCre-ER, B6.Cg-Tg(CAG-cre/Esr1*)5Amc/J; Strain #: 004682) (*Hayashi and McMahon, 2002*), transgenic *Sox2-Cre* (B6.Cg-Edil3Tg(*Sox2*-cre)1Amc/J; Strain #: 008454) (*Hayashi et al., 2002*), and transgenic *Nestin-Cre* (B6.Cg-Tg(Nes-cre)1Kln/J; Strain #: 003771)

(*Tronche et al., 1999*), and *Vav-iCre* transgenic mice (B6.Cg-*Commd10*[Tg(Vav1-icre)A2Kio]/J; Strain #:008610) (*de Boer et al., 2003*) mouse strains were obtained from Jackson Laboratory (*Guo et al., 2022*). The *L3mbt3* deletion mutant (*MBT-1-/+*, B6;129-L3mbtl3tm1Tmiy) mouse strain was previously described (*Arai and Miyazaki, 2005*; *Guo et al., 2022*; *Leng et al., 2018*). The *Dcaf5* deletion mutant mouse strain was produced with gRNA1: CTAGTTAGGTACAATAGGGC and gRNA2: TATTCCTCTGCGACCA CTCA, flanking the exon4 of the mouse *Dcaf5* locus with the altered read-frame in the downstream of protein sequence, in Centre for Phenogenomics (Toronto, Canada) (*Kim et al., 2014a*; *Kleinstiver et al., 2016*; *Slaymaker et al., 2016*). The null *Dcaf5* mutant mice are alive and initially bred with wild-type mice for more than 10 generations to ensure the knock-out effects. The *Ezh2*[K20R] knock-in mice were produced with the gRNA: ACACGCTTCCGCCAACAAAC and the repair template of a single-strand oligonucleotide with the nucleotide changes encoding c.59_60AA >GG required for the K20R change of mouse *Ezh2* at Phenogenomics (*Kim et al., 2014a*; *Kleinstiver et al., 2016*; *Slaymaker et al., 2016*). The mouse *L3mbtl3*[tm1a(EUCOMM)Hmgu] embryonic stem cells containing the verified conditional LoxP sites flanking the exon 5 of *L3mbtl3* were obtained from European Mouse Mutant Cell Repository (EuMMCR) and the *L3mbtl3tm1a(EUCOMM)Hmgu* mice were produced at University of California, Davis/KOMP Repository. The *L3mbtl3tm1a(EUCOMM)Hmgu* mice were bred with the FLPo-10 mouse strain (B6.Cg-Tg(Pgk1-flpo)10Sykr/J; Strain #: 011065) from Jackson Laboratory to delete the LacZ and Neo cassettes to establish the *L3mbtl3*[fl/+] conditional mutant mice (*Wu et al., 2009*). All the mutant mice were DNA sequenced and verified. All animal experiments, including breeding, housing, genotyping, and sample collection, were conducted in accordance with the animal protocols approved by the Institutional Animal Use and Care Committee (IACUC) and complied with all relevant ethical regulations at the University of Nevada, Las Vegas, with the IACUC approved project numbers (IACUC-01161)711621 and (IACUC-01177)832146. All procedures were conducted according to the National Institutes of Health (NIH) Guide for Care and Use of Laboratory Animals. The UNLV IACUC is an AAALAC-approved facility and meets the NIH Guide for the Care and Use of Animals.

## Animal phenotype analysis

For mouse embryo analyses, usually 3 pairs of the heterozygous (-/+) male and female mice (10–12 weeks old) in three cages, each with 1 male and 1 female, were bred in the late afternoon and the breeding plugs were examined in the female mice in next morning (*Guo et al., 2022*). The positive plugs were counted as embryonic day 1 (E1) and the pregnant female mice between E14-E17.5 were euthanized by the primary method of $CO_2$ asphyxiation, followed by cervical dislocation (secondary method), as approved by the institutional IACUC committee (*Guo et al., 2022*). Usually, a single pregnant female mouse produced about 6–8 embryos, which segregated at the Mendelian inheritance ratio, usually with 1–2 *L3mbt3*[-/-] or *Dcaf5*[-/-], 1–2 wild-type, and 3–4 heterozygous *L3mbtl3*[-/+] or *Dcaf5*[-/+] embryos. The *L3mbtl3* null embryos between E17.5–19.5 usually died and became disintegrated so they were excluded from protein analyses (*Arai and Miyazaki, 2005*; *Guo et al., 2022*; *Leng et al., 2018*). For the analysis of PRC2 proteins in *Kdm1a*[fl/fl]/*Nestin-Cre* mice, usually, 3–4 pairs of the *Kdm1a*[fl/fl] male and *Kdm1a*[fl/+]/*Nestin-Cre* female mice (10–12 weeks old) were bred (*Guo et al., 2022*). The animals were collected immediately after birth to avoid any delay in sample analysis. The brains of the mice were dissected for protein or immunostaining analysis. For immunostaining, embryos or dissected brains were fixed in 4% paraformaldehyde (PFA) at 4 °C overnight and embedded in optimal cutting temperature compound (O.C.T) according to standard procedures (*Christopher et al., 2017*; *Guo et al., 2022*). Sections (10 µm thick, coronal) were stained with specific antibodies and counter-stained with 4',6-diamidino-2-phenylindole (DAPI) (*Guo et al., 2022*). Images were acquired with the Nikon A1Rsi Confocal LSM. The sample size was chosen on the basis of our experience on *L3mbtl3, Dcaf5, Kdm1a, or Ezh2*[K20R] mutant mice and on cultured cells in order to detect the EZH2 and H3K27me3 proteins for differences of at least 50% between the wild-type and mutant groups (*Guo et al., 2022*). For analysis of *Ezh2*[K20R] mice, the heterozygous *Ezh2*[K20R/+] mice were bred to obtain the wild-type, *Ezh2*[K20R/+] heterozygous, and *Ezh2*[K20R/K20R] homozygous mice, as the *Ezh2*[K20R/K20R] homozygous mutants survive. In the experimental analyses for the examination of proteins, the investigators were unaware of the genotypes of the experimental embryos. The investigators also randomly analyzed the wild-type, heterozygous and homozygous knockdown embryos. For the analysis of proteins and DNA from embryos, the experimental procedures for embryo isolation were approved by the UNLV Institutional

Animal Use and Care Committee (IACUC). The embryos from the euthanized pregnant female mice or dissected brains, spleens, or livers from the conditional knockout mice, washed with PBS, and lysed in the NP40 lysis buffer (*Guo et al., 2022*). The nuclear and cytosolic fractions were separated by centrifugation. Genomic DNA was isolated from nuclear pellets by Zymo genomic DNA-tissue prep kit and quantified. Proteins in the cytosolic supernatant of the lysates were quantified by protein assay dye (Bio-Rad), equalized, and boiled for 15 min after addition of 1% SDS and 5% beta-mercaptoethanol to the lysates. Proteins were resolved in protein gel and analyzed by Western blotting (*Guo et al., 2022*).

## Flow cytometry

Flow cytometry analyses were performed using a SONY SH800 high-speed multilaser flow cytometer and cell sorter with the FlowJo software in the Core Facility of Nevada Institute of Personalized Medicine. Single-cell suspensions were harvested from bone marrow and lysed with the ACK buffer (ThermoFisher). For mature cells, cells were analyzed with directly conjugated anti-mouse antibodies (Biolegend): Ly-6G-PE (1A8), Gr-1-PE (RB6-8C5), CD11b/Mac1-APC (M1/70), CD4-PE/Cyanine7 (GK1.5), CD8a-APC (53-6-7), and B220-PE (RA3-6B2)(*Akashi et al., 2000*; *Arai and Miyazaki, 2005*; *Traver et al., 2001*). For immature cells, depletion of lineage cells was labeled with a cocktail consisting of biotinylated antibodies (Biolegend): Gr-1 (RB6-8C5), TER-119, CD3e (145–2 C11), and CD11b (M1/70), conjugated to the EasySep Mouse Streptavidin RapidSpheres (Cat: #19860 A, STEMCELL Technologies), and separated on the EasySep Magnet (Cat:#18000) according to the accompanying protocol from STEMCELL Technologies (*Akashi et al., 2000*; *Arai and Miyazaki, 2005*; *Traver et al., 2001*). For immature cells, directly conjugated antibodies used were as follows: streptavidin-FITC (Cat: 405201), c-Kit-PE/Cyanine7 (2B8), Sca-1-PE/Dazzle 594 (D7), CD16/CD32-PE (2.4G2), and CD34 Alexa Fluor 647 (RAM34). Dead cells were stained with Zombie Green (Cat: 423111)(*Akashi et al., 2000*; *Arai and Miyazaki, 2005*; *Traver et al., 2001*). All these antibodies were used at 1:150.

## RNA extraction and qRT-PCR analysis

RNA was extracted from the heads of mouse embryos or neonatal mice using Trizol regent (Thermo Fisher) according to the manufacturer's instructions (*Guo et al., 2022*). 1 μg of total RNA was reverse-transcribed using a first-strand cDNA synthesis kit (Invitrogen). qRT-PCR assays were performed with SYBR Green Mastermix (Bio-Rad) and specific primers for PCR amplification. qRT-PCR data were recorded and analyzed using iQ-PCR (Bio-Rad) equipment and software according to manufacturers' recommendations (*Guo et al., 2022*). For each primer pair, the primer efficiency was measured and the melting curve was analyzed. For each experiment, three technical replicates were used. The primers used for the qRT-PCR studies are in *Supplementary file 1*.

## Statistical information

Experiments were usually performed with at least three independent repeats (biological replicates) to ensure the results (*Guo et al., 2022*; *Leng et al., 2018*). For animal experiments, triplicated breeding was used to obtain a statistically significant number of embryos or mice; and statistically significant differences between means of protein levels in the control wild-type and knockout mutants were compared using a two-tailed equal-variance independent Student's t-test (*Guo et al., 2022*). All other data were determined using a two-tailed equal-variance independent Student's t-test (*Guo et al., 2022*). The data in all figures met the assumption of normal distribution for tests. Different data sets were considered to be statistically significant when the p-value was <0.05 (*), 0.01 (**), 0.001 (***), or 0.0001 (****)(*Fay and Gerow, 2013*).

# Acknowledgements

This work was supported by grants from the National Institutes of Health (R15CA254827 to HS and R01GM140185 to HZ). The DNA sequencing analysis of animal mutations was supported by the Nevada INBRE Scientific Core Service Award. The flow cytometry analysis was conducted in the Core Facility of the Nevada Institute of Personalized Medicine.

## Additional information

### Funding

| Funder | Grant reference number | Author |
|---|---|---|
| National Institutes of Health | R15CA254827 | Hong Sun |
| National Institutes of Health | R01GM140185 | Hui Zhang |

The funders had no role in study design, data collection and interpretation, or the decision to submit the work for publication.

### Author contributions

Pengfei Guo, Data curation, Formal analysis, Investigation, Methodology; Rebecca C Lim, Keshari Rajawasam, Investigation, Methodology; Tiffany Trinh, Methodology; Hong Sun, Conceptualization, Formal analysis, Supervision, Funding acquisition, Investigation, Project administration; Hui Zhang, Conceptualization, Supervision, Funding acquisition, Investigation, Methodology, Writing - original draft, Project administration, Writing - review and editing

### Author ORCIDs

Pengfei Guo ⓘ http://orcid.org/0000-0001-6734-6549
Hui Zhang ⓘ http://orcid.org/0000-0001-6028-2554

### Ethics

All animal experiments including breeding, housing, genotyping, and sample collection were conducted in accordance with the animal protocols approved by the institutional Animal Use and Care Committee (IACUC) and complied with all relevant ethical regulations at the University of Nevada, Las Vegas. All procedures were conducted according to the National Institutes of Health (NIH) Guide for Care and Use of Laboratory Animals. The UNLV IACUC is an AAALAC approved facility and meets the NIH Guide for the Care and Use of Animals. protocols (IACUC-01161)711621 and (IACUC-01177)832146 described previously (Guo et al., 2022).

### Decision letter and Author response

Decision letter https://doi.org/10.7554/eLife.86168.sa1
Author response https://doi.org/10.7554/eLife.86168.sa2

## Additional files

### Supplementary files

- MDAR checklist
- Supplementary file 1. Final list of oliognucleotide primers for RT-PCR.

### Data availability

All data generated during this study are included in the manuscript. Uncropped immunoblots, immunostaining, and gel blot images are accessible as source data.

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

# Appendix 1

### Appendix 1—key resources table

| Reagent type (species) or resource | Designation | Source or reference | Identifiers | Additional information |
|---|---|---|---|---|
| Gene (*Homo sapiens*) | EZH2 | RefSeq | 2146, NM_001203247.2 | DNA sequencing authentication. |
| Strain (*Mus musculus*) | Kdm1a$^{fl/+}$conditional mutant strain | Jackson Laboratory | B6.129-Kdm1a tm1.1Sho/J; Strain #: 023969 | *Kerenyi et al., 2013* |
| Strain (*Mus musculus*) | Transgenic actin-Cre-ER strain | Jackson Laboratory | CAGGCre-ER, B6.Cg-Tg(CAG-cre/Esr1*)5Amc/J; Strain #: 004682 | *Hayashi and McMahon, 2002* |
| Strain (*Mus musculus*) | Transgenic Vav-iCre transgenic mice | Jackson Laboratory | B6.Cg-Commd10$^{Tg(Vav1-icre)A2Kio}$/J; Strain #:008610 | *de Boer et al., 2003* |
| Strain (*Mus musculus*) | transgenic Sox2-Cre strain | Jackson Laboratory | B6.Cg-Edil3Tg(Sox2-cre)1Amc/J; Strain #: 008454 | *Hayashi et al., 2002* |
| Strain (*Mus musculus*) | L3mbt3+/-mutant strain | *Leng et al., 2018*, | Mbt-1+/-, B6;129-L3mbtl3tm1Tmiy | *Arai and Miyazaki, 2005* |
| Strain (*Mus musculus*) | Dcaf5+/-mutant strain | This paper | Centre for Phenogenomics (Toronto, Canada) | produced with gRNA1: CTAG TTAGGTACAATAGGGC and gRNA2: TATTCCTCTGCG ACCACTCA. |
| Strain (*Mus musculus*) | L3mbtl3$^{fl/+}$ mutant strain | European Mouse Mutant Cell Repository (EuMMCR) | L3mbtl3tm1a(EUCOMM)Hmgu mice | Produced with the FLPo-10 mouse strain. |
| Strain (*Mus musculus*) | FLPo-10 mouse strain | Jackson Laboratory | B6.Cg-Tg(Pgk1-flpo)10Sykr/J; Strain #: 011065 | *Wu et al., 2009* |
| Cell line (*Homo sapiens*) | 293T | ATCC | CRL-3216 | Authenticated by high levels of CDK inhibitor CDKN2A and p53 |
| Cell line (*Homo sapiens*) | HCT116 | ATCC | CCL-247 | Authenticated by expression of wild-type p53 and induction of CDKN1A by UV irradiation |
| Cell line (*Homo sapiens*) | G401 | CRL-1441 | ATCC | Authenticated by lack of expression of SMARCB1 |
| Cell line (*Homo sapiens*) | T47D | ATCC | HTB-133 | Authenticated by lack of ARID1A expression |
| Cell line (*Homo sapiens*) | HeLa | ATCC | CRM-CCL-2 | Authenticated by high levels of CDK inhibitor CDKN2A and p53 |
| Cell line (*Homo sapiens*) | PA-1 | ATCC | CRL-1572 | Authenticated by expression of SOX2 and OCT4 |
| Cell line (*Homo sapiens*) | H1299 (NCI-H1299) | ATCC | CRL-5803 | Authenticated by lack of p53 |
| Cell line (*Homo sapiens*) | H520 (NCI-H520) | ATCC | HTB-182 | Authenticated by high expression of SOX2 |

*Appendix 1 Continued on next page*

*Appendix 1 Continued*

| Reagent type (species) or resource | Designation | Source or reference | Identifiers | Additional information |
|---|---|---|---|---|
| Cell line (*Mus musculus*) | Mouse embryonic fibroblasts (MEFs) | This paper | Primary embryonic fibroblasts from isolated mouse embryos | Primary cells; prepared according to IACUC approved protocols (IACUC-01161) 711621 and (IACUC-01177)832146. |
| Cell line (*Mus musculus*) | Mouse embryonic fibroblasts (MEFs) from K20R mutant mice | This paper | Mouse embryonic fibroblasts (MEFs) from homozygous Ezh2$^{K20R/K20R}$ mutant mice | Primary cells; prepared according to IACUC approved protocols (IACUC-01161) 711621 and (IACUC-01177)832146. |
| Cell line (*Mus musculus*) | L3mbtl3-knockout MEFs | This paper | MEFs from homozygous L3mbtl3 KO mutant mice | Primary cells; prepared according to IACUC approved protocols (IACUC-01161)711621 and (IACUC-01177)832146. |
| Cell line (*Mus musculus*) | Kdm1a$^{fl/fl}$ MEFs | This paper | MEFs from homozygous Kdm1a$^{fl/fl}$ mutant mice | Primary cells; prepared according to IACUC approved protocols (IACUC-01161) 711621 and (IACUC-01177)832146. |
| Cell line (*Mus musculus*) | L3mbtl3$^{tm1a(EUCOMM)Hmgu}$ (L3mbtl3$^{fl/fl}$) | This paper | MEFs from homozygous (L3mbtl3$^{fl/fl}$) mice | Primary cells; prepared according to IACUC approved protocols (IACUC-01161) 711621 and (IACUC-01177)832146. |
| Cell line (*Mus musculus*) | Kdm1a$^{fl/fl}$/ L3mbtl3$^{fl/fl}$/ actin-Cre-ER | This paper | Kdm1a$^{fl/fl}$/ L3mbtl3$^{fl/fl}$/ actin-Cre-ER | Primary cells; prepared according to IACUC approved protocols (IACUC-01161) 711621 and (IACUC-01177)832146. |
| Transfected construct (human) | Kdm1a siRNA #1 | Synthesized from Horizon Discovery | *Guo et al., 2022* | transfected construct (human) |
| Transfected construct (human) | Kdm1a siRNA #2 | Synthesized from Horizon Discovery | *Guo et al., 2022* | transfected construct (human) |
| Transfected construct (human) | Kdm1a-3'UTR siRNA | Synthesized from Horizon Discovery | *Guo et al., 2022* | transfected construct (human) |
| Transfected construct (human) | Dcaf5-1 siRNA #1 | Synthesized from Horizon Discovery | *Guo et al., 2022* | transfected construct (human) |
| Transfected construct (human) | Dcaf5-2 siRNA #1 | Synthesized from Horizon Discovery | *Guo et al., 2022* | transfected construct (human) |
| Transfected construct (human) | L3mbtl3-1 siRNA #1 | Synthesized from Horizon Discovery | *Guo et al., 2022* | transfected construct (human) |
| Transfected construct (human) | L3mbtl3-2 siRNA #1 | Synthesized from Horizon Discovery | *Guo et al., 2022* | transfected construct (human) |
| Transfected construct (human) | Set7-1 siRNA #1 | Synthesized from Horizon Discovery | *Guo et al., 2022* | transfected construct (human) |
| Transfected construct (human) | Set7-2 SMART pool | Synthesized from Horizon Discovery | *Guo et al., 2022* | transfected construct (human) |
| Antibody | Anti-KDM1A antibody | Fortis Life Sciences | A300-215A | IF(1:1000), WB (1:1000) |
| Antibody | anti-L3MBTL3 antibody | Fortis Life Sciences | A302-852 | IF(1:1000), WB (1:1000) |

*Appendix 1 Continued on next page*

*Appendix 1 Continued*

| Reagent type (species) or resource | Designation | Source or reference | Identifiers | Additional information |
|---|---|---|---|---|
| Antibody | anti-SUZ12 antibody | Fortis Life Sciences | A302-407A | IF(1:1000), WB (1:1000) |
| Antibody | anti-SET7 antibody | Fortis Life Sciences | A301-747A | IF(1:1000), WB (1:1000) |
| Antibody | Anti-EED antibody | Abcam | ab236292 | IF(1:1000), WB (1:1000) |
| Antibody | Anti-Phospho-EZH2 (S21) | Affinity Biosciences | AF3822 | IF(1:1000), WB (1:1000) |
| Antibody | Anti-K20me antibody | This paper | Affinity purified Anti-K20me antibody | IF(1:1000), WB (1:1000) |
| Antibody | anti-GFI1B antibody | Cell Signaling Technology | 5849 | IF(1:1000), WB (1:1000) |
| Antibody | Anti-EZH2 antibody | Cell Signaling Technology | 5246 | IF(1:1000), WB (1:1000) |
| Antibody | anti-H3K27me3 antibody | Cell Signaling Technology | 9733 | IF(1:1000), WB (1:1000) |
| Antibody | Anti-Actin antibody | Santa Cruz Biotechnologies | Sc-1616 | WB (1:5000) |
| Antibody | Anti-FLAG M2 antibody | Sigma | F1804 | WB (1:5000) |
| Antibody | ant-HA antibody | Sigma | 11867423001 | WB (1:5000) |
| Antibody | anti-GFP antibody | Sigma | 11814460001 | WB (1:5000) |
| Antibody | Anti-GAPDH | Proteintech | 60004–1-Ig | WB (1:5000) |
| Antibody | Rabbit anti-L3MBTL3 | This paper | *Guo et al., 2022* | IP(1:1000), WB (1:1000) |
| Antibody | Anti-DCAF5 antibody | This paper | *Guo et al., 2022* | IP(1:1000), WB (1:1000) |
| Commercial assay or kit | Lipofectamine 2000 Transfection Reagent | Thermo Fisher Scientific | 11668019 | |
| Commercial assay or kit | Oligofectamine Regent | Thermo Fisher Scientific | 2399123 | |
| Commercial assay or kit | DharmaFECT 1 Transfection Reagent | Horizon Discovery | T-2001–03 | |
| Commercial assay or kit | TRIzol reagent | Life Technologies | 423707 | |
| Commercial assay or kit | SuperScript III First-Strand Synthesis System for RT-PCR | Life Technologies | 2490151 | |
| Commercial assay or kit | E.Z.N.A TISSUE DNA Kit | Omega | d3396-02 | |
| Commercial assay or kit | Sulfolinkcoupled-resins | ThermoFisher Scientific | XC339981 | |
| Recombinant DNA reagent | pMSCV--Puro vector | Clontech | 634401 | |
| Recombinant DNA reagent | pEGFP-C1 | Clontech | 6084–1 | |
| Recombinant DNA reagent | pCDNA3.1-puro vector | Invitrogen | Size: 5446 NT | |
| Recombinant DNA reagent | pKH3-vector | Addgene | 12555 | |

