## [Editor Report]

This is a valuable study elucidating a novel mechanism of EZH2 regulation. The evidence supporting the claims of the authors is solid, with the inclusion of the large number of data obtained from animal models. This study is of general interest to audiences in the epigenetics field.

---

## [Decision Letter]

**Decision letter after peer review:**

Thank you for submitting your article "A Methylation-Phosphorylation Switch Controls EZH2 Stability and Hematopoiesis" for consideration by *eLife*. Your article has been reviewed by 3 peer reviewers, one of whom is a member of our Board of Reviewing Editors, and the evaluation has been overseen by Kevin Struhl as the Senior Editor. The following individual involved in the review of your submission has agreed to reveal their identity: Qi Cao (Reviewer #2).

Essential revisions:

This study shows that SET7 and LSD1 regulates the dynamic methylation of EZH2 at K20, which is recognized by L3MBTL3 promoting protein degradation via the DCAF5-CRL4 E3 ubiquitin ligase. K20 methylation negatively regulates S21 phosphorylation and vice versa, modulating EZH2 functions. Mice harboring the K20 methylation-deficient mutant (K20R) exhibit hematopoietic defects and reactive hyperplasia. Overall, this is an interesting study elucidating a novel mechanism of EZH2 regulation. Methodologies are sound and the conclusions are largely supported by the data provided. However, there are some questions regarding the overall model and some contradictory results.

1. The overall model is that SET7-mediated EZH2K20 methylation promotes EZH2 protein degradation, which suggests that this mark negatively regulates EZH2. However, Figure 4D-F (and Figure 5D) show that EZH2 protein level does not change upon overexpression of SET7, and in contrast, the global H3K27m3 level increases, indicating an overall positive role. Although the authors attribute this to the decrease of EZH2 S21 phosphorylation, it seems paradoxical that a modification positively and negatively regulates the target protein simultaneously. If the main function of K20me is to promote EZH2 proteolysis, why do cells also utilize it to activate EZH2 enzymatic activity?

2. The effect of LSD1 KO on EZH2 is drastic, leading to almost diminished levels of EZH2 and H3K27me3 (Figures 1, 2, 5f, etc), suggesting that the majority of EZH2 in cells is methylated. Quantitative MS need to be performed to assess the K20 methylation levels in cells and under different treatments (e.g., +/-LSD1 KO, +/-SETD7 OE, and +/- MG132 etc). More importantly, based on the proposed model of K20me-S21phos crosstalk, one would expect an increase in H3K27me3 level upon LSD1 KO, as seen above in SET7 overexpression. The seemingly contradictory results of SET7 OE and LSD1 KO need to be discussed.

3. Figure 6 is the main evidence to support the conclusion that the methylation-phosphorylation switch regulates the stability of EZH2, but only the K20me and S21p of EZH2 in MEFs is presented in Figure 6B. How about the K20me and S21p levels in T47D and H1299 cells after MK2206 treatment? Quantitative MS should also be done for in these cells expressing EZH2-K20R and S21A mutants. This are critical experiments to demonstrate the feedback regulation between K20 methylation and S21 phosphorylation.

4. The authors propose that the levels of L3MBTL3 in cells determine the fate of EZH2K20me: high levels of cellular L3MBTL3 promote EZH2 degradation and low levels of L3MBTL3 lead to hyperactive EZH2. If this is the case, one would expect to see a negative correlation of EZH2 and L3MBTL3 at the protein level across cell lines. Based on Figure 6-Suppl Figure 2, T47D belongs to the low L3MBTL3 cell lines. However, LSD1 knockdown in T47D can still downregulate EZH2 protein. This seems to be contradictory to their hypothesis. Similar experiments need to be done in some other L3MBTL3-low cell lines. Furthermore, It seems that low expression or alterations of L3MBTL3 are not rare in cancer cells. Further discussion would be helpful about how broadly the methylation-phosphorylation switch controls EZH2 stability during development and disease initiation/progression.

5. There are no data directly demonstrating that the enzymatic activities of LSD1 and SET7 are required for EZH2 regulation Rescue experiments using WT and enzymatic dead mutants in the KD or KO cells are necessary.

6. Given that methylation-phosphorylation switch of EZH2 is likely universal, it is interesting that K20R GEMM developed hematopoiesis. Do the mouse models of Lsd1 KO, L3mbtl3 KO, or Dcaf5 KO also develop hematopoiesis? There is a global increase of H3K27me3 in the K20R-expressing mice, however, it is surprising that many genes such as GFI1B are upregulated. Is it through a H3K27me3-independent function of EZH2? The authors should also evaluate the H3K27me3 levels and expression of the classical EZH2 target genes to see if they are downregulated in K20R-expressing cells.

*Reviewer #1 (Recommendations for the authors):*

1. The overall model is that SET7-mediated EZH2K20 methylation promotes EZH2 protein degradation, which suggests that this mark negatively regulates EZH2. However, Figure 4F (and Figure 5D) shows that the EZH2 protein level does not change upon overexpression of SET7, and in contrast, the global H3K27m3 level increases, indicating an overall positive role. Although the authors attribute this to the decrease of EZH2 S21 phosphorylation, it seems paradoxical that a modification positively and negatively regulates the modified protein simultaneously. If the main function of K20me is to promote protein degradation, why do cells also utilize it to activate EZH2 enzymatic activity?

2. The effect of LSD1 KO on EZH2 is drastic, leading to almost diminished levels of EZH2 and H3K27me3 (Figures 1, 2, 5f, etc), suggesting that the majority of EZH2 is methylated. Quantitative MS needs to be performed to assess the K20 methylation levels +/-LSD1 KO and +/- MG132. More importantly, based on the proposed model of K20me-S21phos crosstalk, one would expect an increase in the H3K27me3 level upon LSD1 KO, as seen above in SET7 overexpression.

3. Figure 5F, it is surprising that S21A leads to global loss of H3K27me3, given that the endogenous EZH2 still exists presumably. Endogenous EZH2 and EZH2K20me need to be probed. More importantly, is S21A enzymatically dead and does it function as a dominant negative mutant?

4. Figure 5G and H, overexpression of L3MBTL3 and DCAF5 promotes EZH2 ubiquitination. But why it does not affect global EZH2 levels?

5. Figure 6, the authors propose that L3MBTL3 levels in cells determine the outcome of EZH2K20 methylation. if this is the case, one would expect to see a negative correlation of EZH2 and L3MBTL3 at the protein level across cell lines.

*Reviewer #2 (Recommendations for the authors):*

1. P7, the conclusion for the section "Deletion of mouse L3mbtl3 gene causes the accumulation of EZH2 protein" is inappropriate. The data presented in Figure 2 only demonstrated that L3mbtl3 deletion can rescue LSD1 silencing reduced EZH2 protein levels, unrelated to L3MBTL3-dependent proteolysis.

2. Figure 4 needs to be reorganized to display more logically.

3. Figure 6 is the main evidence to support one of the important conclusions that the methylation-phosphorylation switch regulates the stability of EZH2, but only the K20me and S21p of EZH2 in MEFs were presented in Figure 6B. How about the K20me and S21p levels in T47D and H1299 cells after MK2206 treatment?

4. It seems that the low expression and alterations of L3MBTL3 are not rare in cancer cells. Further discussion is needed about how broadly the methylation-phosphorylation switch controls EZH2 stability during development and disease initiation/progression.

*Reviewer #3 (Recommendations for the authors):*

1. There are no data directly demonstrating that enzymatic activity dead mutants of LSD1, SET7, and L3MBTL3 lose their roles in the regulation of EZH2 methylation and protein stability. Almost all the experiments utilized knockdown or knockout strategies, which cannot exclude the off-target or secondary effects. Rescue experiments may be essential.

2. In Figure 1F, the authors showed that LSD1 knockdown can indeed downregulate the protein levels of EZH2 in T47D. But if L3 is low in T47D, shouldn't they see no effect of LSD1 on EZH2 protein stability?

3. In Figures 2D and 3D, shouldn't we see a global increase in the staining intensities of EZH2 and H3K27me3? It is not sure why the authors highlighted some regions of the coronal sections. If they think only in those regions EZH2 and H3K27me3 levels were changed, please explain the specificity of those regions.

3. Does the antibody recognize di- or tri-methylation of EZH2 at K20 or mono-methylation at other lysine residues? These types of modified peptides may be included in the in vitro dot assay to further prove the specificity of the antibody. It is also unclear if SET7 is the major methyltransferase responsible for the methylation of EZH2 at K20 in cells.

4. In Figure 4D-F, shouldn't manipulation of SET7 change the protein levels of EZH2? Overexpression of SET7 leads to downregulation of EZH2 (but H3K27me3 in Figure 4F was actually upregulated), whereas knockdown of SET7 should stabilize EZH2.

5. In Figure 5B, there is no data showing the "gradual reduction of the S21-phosphorylated form of EZH2". It is interesting to see that phosphorylated AKT was actually reduced during mouse embryonic development. Why is that?

6. The converged effect of EZH2 K20 methylation and S21 phosphorylation on H3K27me3 is confusing. K20 methylation of EZH2 destabilizes the histone methyltransferase, whereas S21 phosphorylation impairs its enzymatic activity. However, K20 mono-methylation prevents S21 phosphorylation. Then which modification will win over in terms of deciding the H3K27me3 levels? And why?

7. The conflicting result that the methylation-phosphorylation switch of EZH2 is defective in T47D cells is very confusing. If this is due to the low level of L3MBTL3, why LSD1 knockdown in T47D can still downregulate EZH2 protein then (Figure 1)? Although the authors showed in Supplemental Figure 2 that L3MBTL3 levels are various in different cell lines, they didn't really show whether in those L3MBTL3-low cells other than T47D, they see the same results as in T47D.

8. The most direct data to demonstrate the negative feedback between K20 methylation and S21 phosphorylation is to detect the levels of these two modifications in cells expressing wild-type EZH2 or K20R or S21A mutant using the specific antibodies detecting the modified EZH2. They can also overexpress SET7 or AKT when they express the corresponding mutant.

9. It seems that the methylation-phosphorylation switch of EZH2 is not specific to cancer (observed in MEF) nor any specific types of cancer. Why does K20R overexpression only induce hematopoiesis in the GEMM model? Did the authors see similar results in Lsd1/L3mbtl3/Dcaf5-knockout mice?

10. If H3K27me3 is increased in K20R-expressing mice, why is GFI1B expression upregulated? Is it an H3K27me3-independent function of EZH2? How will the authors reconcile the increase in H3K27me3 levels in K20R-expressing MEF with cytoplasmic localization of this mutant form of EZH2 (Supplemental Figure 4)?

11. What about the classical genes that are repressed by H3K27me3? Are they downregulated in K20R-expressing cells?

---

## [Author Response]

Essential revisions:This study shows that SET7 and LSD1 regulates the dynamic methylation of EZH2 at K20, which is recognized by L3MBTL3 promoting protein degradation via the DCAF5-CRL4 E3 ubiquitin ligase. K20 methylation negatively regulates S21 phosphorylation and vice versa, modulating EZH2 functions. Mice harboring the K20 methylation-deficient mutant (K20R) exhibit hematopoietic defects and reactive hyperplasia. Overall, this is an interesting study elucidating a novel mechanism of EZH2 regulation. Methodologies are sound and the conclusions are largely supported by the data provided. However, there are some questions regarding the overall model and some contradictory results.1. The overall model is that SET7-mediated EZH2K20 methylation promotes EZH2 protein degradation, which suggests that this mark negatively regulates EZH2. However, Figure 4D-F (and Figure 5D) show that EZH2 protein level does not change upon overexpression of SET7, and in contrast, the global H3K27m3 level increases, indicating an overall positive role. Although the authors attribute this to the decrease of EZH2 S21 phosphorylation, it seems paradoxical that a modification positively and negatively regulates the target protein simultaneously. If the main function of K20me is to promote EZH2 proteolysis, why do cells also utilize it to activate EZH2 enzymatic activity?

We thank the reviewer’s excellent questions. The original Figure 4D-4E were conducted in H1299 cells stably expressing the Flag-tagged EZH2. In these cells, we showed that siRNA-mediated silencing of LSD1 caused the reduction of the Flag-EZH2 protein, whereas co-silencing of LSD1 and SET7 prevented the reduction of Flag-EZH2 in LSD1 deficient cells. The key question for the data here is why the EZH2 protein level did not decrease or increase when SET7 is overexpressed or silenced, respectively? Most of our repeated silencing experiments for SET7 or L3MBTL3 only showed that they re-stabilize EZH2 protein in LSD1 silenced H1299 cells. Only in a few limited cases, EZH2 protein is stabilized when SET7 or L3MBTL3 is silenced. We would suggest that only a limited fraction of EZH2 is K20 methylated in H1299 cells to be targeted for proteolysis by L3MBTL3 under normal conditions. Another possibility is that L3MBTL3 expression is quite limited in H1299 cells (Figure 6 figure supplement 2A). In addition, the available K20-methylated EZH2 fraction is also affected by other factors, such as the amount of PHF20L1, which binds to methylated lysine residues in proteins. We have previously shown that PHF20L1 binds to the methylated K42 of *Sox2* (JBC, 294 (2), 476-489, 2019) to prevent the degradation of methylated *Sox2* by L3MBTL3 and CRL4^DCAF5^-mediated proteolysis. Our studies suggest that the accumulation or reduction of various methylated substrates to the levels of SET7 or L3MBTK3 is quite complex in various cells, depending on the relative levels of SET7, LSD1, PHF20L1 and L3MBTL3 in the cells. In response to reviewer comments, we have examined the effects of silencing PHF20L1 on EZH2 and found that similar to *Sox2*, loss of PHF20L1 caused the downregulation of EZH2, which can be rescued by L3MBTL3 silencing (new Figure 4—figure supplement 2B and 2C in revision). In addition, we have conducted new experiments showing that ectopic expression of SET7 in human colorectal carcinoma HCT116 cells led to the reduction of both EZH2 protein and H3K27me3 (new Figure 4G in revision). Notably, we usually found that the deletion effects of LSD1 and L3MBTL3 are most pronounced in mouse embryos and in mouse embryonic stem cells (JBC, 294 (2), 476-489, 2019, and Nature Communications volume 13, Article number: 6696, 2022, https://www.nature.com/articles/s41467-022-34348-9), but their loss is less effective in many cultured cancer cell lines. While we are still investigating the mechanistic difference between embryonic stem cells and somatic cancer cells, it seems that the fraction of methylated K20 protein is relatively small and/or L3MBTL3 expression is relatively low in lung carcinoma H1299 cells so that the loss of SET7 often did not induce elevated EZH2 protein levels. The reason for using H1299 cells is because we can establish stable cells that ectopically express the Flag-tagged EZH2 relatively easily in H1299 cells using the retroviral expression system.

For T47D ductal breast carcinoma cells in original Figure 4F, we found it does not respond to SET7 expression. Since the K20-methylated EZH2 requires L3MBTL3, and the L3MBTL3 levels in T47D cells are relatively low, as compared with other cell lines Figure 6—figure supplement 2A, it is likely that even SET7 can methylate K20, but it is not targeted for L3MBTL3 dependent proteolysis due to the low level of L3MBTL3 in T47D cells. We have re-organized this figure to Figure 4—figure supplement 2D in the revision.

In summary, our studies revealed that the activities or levels of SET7 and L3MBTL3 are altered in various cancer cells, so the effects of LSD1 silencing, SET7 ectopic expression, and L3MBTL3 silencing produced different responses in different cancer cells. ON the other hand, all our data are consistent with the hypothesis that K20 methylation targets EZH2 for proteolysis and S21 phosphorylation negatively regulates the methylation dependent EZH2 degradation.

2. The effect of LSD1 KO on EZH2 is drastic, leading to almost diminished levels of EZH2 and H3K27me3 (Figures 1, 2, 5f, etc), suggesting that the majority of EZH2 in cells is methylated. Quantitative MS need to be performed to assess the K20 methylation levels in cells and under different treatments (e.g., +/-LSD1 KO, +/-SETD7 OE, and +/- MG132 etc). More importantly, based on the proposed model of K20me-S21phos crosstalk, one would expect an increase in H3K27me3 level upon LSD1 KO, as seen above in SET7 overexpression. The seemingly contradictory results of SET7 OE and LSD1 KO need to be discussed.

We appreciate the reviewer’s comments and suggestions. We agree that the mouse embryos/neonatal samples would provide a desirable methylated K20 analysis for EZH2. In response to reviewer comments, we initially tried in last April to use the embryos of double Lsd1^fl/fl^ and L3mbtl3^fl/fl^ conditional KO animals with the vav-iCre mediated deletion. We prepared the fetal livers of the wildtype and double KO mutant embryos at embryonic day 18.5-19 (E18.5-19.5). We also prepared the embryonic brains of the wildtype and the L3mbtl3 conditional KO mutant with *Sox2*-cre mediated deletion of L3MBTL3 in the brains from the *Sox2*-cre; L3mbtl3^fl/fl^ mutant embryos at E15.5-16.5. The embryos were lysed in NP40-containing buffer and the EZH2 complex were immunoprecipitated (we purchased several EZH2 antibodies, measured and quantified their immunoprecipitation activities, and picked up the best antibodies for immunoprecipitation with large amount of embryonic lysates). The proteins in the anti-EZH2 IP were separated in protein gels and were silver staining. The EZH2 bands were cut out and sent for mass spectrometry sequencing using the Thermo Scientific Orbitrap Eclips mass spectrometry with ETD, coupled to a Thermo ultimate 3000 nano-LC system in Nevada Proteomics Center at University of Nevada, Reno (UNR). Since the amino-terminal peptide of EZH2 containing the K20 methylation motif is MGQTGKKSEKGPVCWRKRVKSEYMRLRQLKRFRRADEVKTMFSS, trypsin will cut lysine/arginine residues in EZH2 but the RVK peptide will not be easily obtained. We requested to use chymotrypsin that cuts bulky amino acid residues to digest EZH2 protein. The chymotrypsin is not often used and the people in Nevada Proteomics Center did not have much experience of using it. The chymotrypsin peptides were separated by liquid chromatography and sent for MS analysis. Although our silver staining of the EZH2 IP clearly showed EZH2 protein bands, the proteomic facility cannot identify any chymotryptic EZH2 peptides in the MS. When the data quality was lowered, UNR could only detect one peptide match to the EZH2 protein from the sample of the double floxed LSD1 and L3MBTL3 conditional (LSD1^fl/fl^; L3mbtl3^fl/fl^) KO embryos with vav-iCre deletion.

We tried the second MS experiments in last May, with 2 embryonic livers from the double conditional KO mutant mice (vav-cre; LSD1 flox/flox; L3 flox/flox), and that of the wildtype animals at E16.5. We also used the embryonic brain of floxed L3mbtl3 (L3mbtl3fl/fl;*Sox2*-cre) with *Sox2*-cre, and that of the wildtype animals at E16.5. Again, the silver stained EZH2 bands from anti-EZH2 IPs were clearly detected and these protein bands were digested with chymotrypsin. But this set of experiments using the MS analysis of the chymotrypsin digested peptides again did not produce any peptide match with EZH2. The UNR Proteomics facility suggested to use trypsin digestion. We agreed and the tryptic digestion produced five EZH2 peptide matches but the desired peptides containing either K20 or S21 were not detected.

We tried the third time with more EZH2 proteins. We bred additional conditional L3mbtl3^fl/fl^ conditional knockout mice with *Sox2*-cre (L3mbtl3^fl/fl^;*Sox2*-cre) to obtain 10 mutant embryonic brains with L3mbtl3 deletion in October, 2023. The anti-EZH2 IP samples were stained with Coomassie Brilliant Blue and digested with chymotrypsin. We have obtained the sequences of five EZH2 peptides but the K20/S21 containing peptide was still missing. We believe our EZH2 protein band was in good amount, as show in the following figure. In this figure, band A is the EZH2. This round of chymotrypsin digestion, we successfully obtained about 16% total coverage of the EZH peptides. But the desired peptides containing either K20 or S21 were not among them.

**Author response image 1. sa2fig1:** 

Since the breeding of *Sox2*-cre with double conditional KO of LSD1 and L3MBTL3 (*Sox2*-cre;LSD1^fl/fl^; L3mbtl3^fl/fl^) would take a long breeding time, we would consider that the request to conduct the MS analysis of EZH2 peptides containing K20 methylation and S21 phosphorylation in our research would be very difficult to fulfill with certain amount of time, and it might require a great deal of effort, time, and grant support to characterize the conditions for physically detecting K20 methylation in EZH2. Since the effects of LSD1 deletion is more pronounced in the mice than that of cultured somatic cancer cells, it would be extremely difficult to conduct MS analysis using EZH2 purification for K20 methylation detection in cultured cells, given that the methylated EZH2 is only a relatively small fraction of total EZH2 protein. On the other hand, we would very much like to know whether K20 of EZH2 is mono- or di-methylated. However, since SET7 is reported to monomethylate H3K4 (Nature 421, 652-656, 2003), it is reasonable to suggest that SET7 can monomethylate K20 of EZH2.We apologize for the confusion raised by the reviewers on the effects of LSD1 KO. In the Nestin-cre mediated Lsd1^fl/fl^ KO mouse brain, EZH2 protein level is reduced due to the accumulation of methylated K20 (Figure 1A and 1B), a substrate for LSD1. EZH2 protein level is reduced when LSD1 is deleted in mouse embryonic fibroblasts, resulting in the reduction of the H3K27me3 level (Figure 1D). In human rhabdoid tumor G401 cells, loss of LSD1 also resulted in the degradation of wildtype Flag-EZH2 or endogenous EZH2 proteins, whereas in EZH2 K20R mutant expressing cells, EZH2 K20R protein and H3K27me3 levels were not affected by LSD1 silencing (Figure 5F). However, the S21A mutant of Flag-EZH2 is still sensitive to the loss of LSD1, as S21A mutation may facilitate K20 methylation, promoting the reduction of EZH2 and consequently the lower levels of H3K27me3. We apologize for the confusion in Figure 5F. In addition, to further verify the effect of LSD1 loss in cultured cells, we have conducted new experiments using siRNA-mediated knockdown of LSD1 to examine the effects on EZH2 and H3K27me3 levels in HCT116 cells. We found that both EZH2 and H3K27me3 are downregulated after LSD1 silencing, but treatment of LSD1-deficient cells HCT116 cells with the protease inhibitor MG132 restored the levels of both EZH2 and EZH2-k20me (new Figure 4D). We also added new experiments to measure the EZH2-K20me levels in the Lsd1^fl/fl^ conditional KO and L3mbtl3 KO mice. As shown in Figure 4E, LSD1 loss in the Nestin-Cre Lsd1^fl/fl^ conditional KO mice reduced the methylated EZH2 and total EZH2 proteins. Conversely, L3mbtl3 knockout resulted in the increased K20 methylation levels of EZH2 and total EZH2 proteins (Figure 4F). We hope these experiments help clarify the confusion raised by the reviewers.

3. Figure 6 is the main evidence to support the conclusion that the methylation-phosphorylation switch regulates the stability of EZH2, but only the K20me and S21p of EZH2 in MEFs is presented in Figure 6B. How about the K20me and S21p levels in T47D and H1299 cells after MK2206 treatment? Quantitative MS should also be done for in these cells expressing EZH2-K20R and S21A mutants. This are critical experiments to demonstrate the feedback regulation between K20 methylation and S21 phosphorylation.

We thank the reviewer’s comments and agree that Figure 6 is very important to support the methylation-phosphorylation switch model that regulates the stability of EZH2. In response to reviewer’s comments, we conducted new experiments to examine the effects of AKT inhibitor MK2206 on K20 methylation and S21 phosphorylation in human teratocarcinoma PA-1 cells (Figure 6C), T47D, and H1299 cells (Figure 6—figure supplement 2B-2C) by Western blot analysis. Consistent with the response of MEFs, MK2206 treatment increased the K20 methylation levels of EZH2, accompanied by reduction of S21 phosphorylation levels in PA-1, H1299, and T47D cells. MK2206 treatment also caused on the reduction of EZH2 protein and decreased levels of H3K27me3 levels in PA-1 and H1299 cells, the AKT inhibitor had marginal effects on the levels of EZH2 protein and had slightly increased levels of H3K27me3 in T47D, consistent with our original data on T47D cells that T47 cells are quite different in response to MK2206 from that of MEFs, PA-1, and H1299 cells (Figure 6A-6C, Figure 6—figure supplement 2B-2C). Thus, our data in MEFs, PA-1, and H1299 cells support the methylation-phosphorylation switch model, while T47D is an exception due to the fact that this cell line did not response to MK2206 well, likely due to the lower levels of L3MBTL3 (Figure 6—figure supplement 2A). Consistent with the low level of L3MBTL3 in T47D cells, the loss of l3mbtl3 in MEFs produced the similar effect in response to MK2206 (Figure 6E). We could not perform the quantitative MS due to the difficulty we had for the MS detection of K20 methylation and S21 phosphorylation mentioned in response to question #2 above.

4. The authors propose that the levels of L3MBTL3 in cells determine the fate of EZH2K20me: high levels of cellular L3MBTL3 promote EZH2 degradation and low levels of L3MBTL3 lead to hyperactive EZH2. If this is the case, one would expect to see a negative correlation of EZH2 and L3MBTL3 at the protein level across cell lines. Based on Figure 6-Suppl Figure 2, T47D belongs to the low L3MBTL3 cell lines. However, LSD1 knockdown in T47D can still downregulate EZH2 protein. This seems to be contradictory to their hypothesis. Similar experiments need to be done in some other L3MBTL3-low cell lines. Furthermore, It seems that low expression or alterations of L3MBTL3 are not rare in cancer cells. Further discussion would be helpful about how broadly the methylation-phosphorylation switch controls EZH2 stability during development and disease initiation/progression.

We thank the reviewer’s constructive comments. We have conducted many experiments on LSD1- and L3MBTL3-dependent EZH2 degradation. We consistently and repeatedly found that siRNA-mediated silencing of LSD1 caused EZH2 proteolysis in cultured human teratocarcinoma PA-1, lung carcinoma H1299, rhabdoid tumor G401, cervical carcinoma HeLa, colorectal carcinoma HCT116, and lung carcinoma H520 cells and MEFs (Figure 1F-G and Figure 1—figure supplement 1). However, human breast carcinoma T47D cells are quite unique because many LSD1 silencing or MK2206 treatment experiments did not cause the reduction of EZH2 protein and decreased levels of H3K27me3. Silencing of LSD1 only occasionally causes some reduction of EZH2 protein, possibly due to low cell density (but we are not sure because it is hard to reproduce the condition). Similarly, Cha et al. did not find that inhibition of AKT caused the downregulation of EZH2 protein in T47D cells (Science, 310(5746), 306-310, 2005). We found that L3MBTL3 levels are quite low in T47D cells, as compared with other cell lines (Figure 6—figure supplement 2A) and stable ectopic expression L3MBTL3 in T47D cells is sufficient to allow T47D cells to reduce EZH2 protein after MK2206 treatment (Figure 6D). These data suggest that L3MBTL3 is limiting in T47D cells. We thank the reviewers to raise this important issue that the lysine methylation mediated protein degradation might be altered in many cancer cells. We will conduct more future experiments to examine this possibility.

5. There are no data directly demonstrating that the enzymatic activities of LSD1 and SET7 are required for EZH2 regulation Rescue experiments using WT and enzymatic dead mutants in the KD or KO cells are necessary.

We thank the reviewer’s helpful comments again! In response, we have conducted new experiments to silence endogenous LSD1 by using the siRNA against the 3’UTR region (si-LSD1-3’UTR) that led to the downregulation of EZH2 and the induced proteolysis of EZH2 but be rescued by the ectopic expression of a functional wildtype Flag-LSD1 cDNA that does not contain the 3’-UTR, but not by the expression of a catalytically dead mutant LSD1 containing only the amino-terminal 1-531 amino acid residues of the Flag-LSD1 cDNA (new Figure 1-supplement 1B and 1C). These experiments indicate that the wildtype LSD1 is functional to suppress the si-LSD1-3’UTR effects of the endogenous LSD1 on EZH2 protein. We also conducted new experiments using MK2206 treatment in PA-1, T47D, and H1299 cells to show that inhibition of AKT and S21 phosphorylation leads to the increase methylation of K20 in EZH2. In new Figure 5D, we showed that expression of wildtype SET7, but not its catalytically inactive form of mutated SET7 (the H297A mutant), can methylate EZH2 and promote the binding of EZH2 to L3MBTL3. Together, these data in the revised manuscript indicate that LSD1 and SET7 are functioning as a demethylase and a methyltransferase for EZH2.

6. Given that methylation-phosphorylation switch of EZH2 is likely universal, it is interesting that K20R GEMM developed hematopoiesis. Do the mouse models of Lsd1 KO, L3mbtl3 KO, or Dcaf5 KO also develop hematopoiesis? There is a global increase of H3K27me3 in the K20R-expressing mice, however, it is surprising that many genes such as GFI1B are upregulated. Is it through a H3K27me3-independent function of EZH2? The authors should also evaluate the H3K27me3 levels and expression of the classical EZH2 target genes to see if they are downregulated in K20R-expressing cells.

We thank the reviewers for the excellent questions. In response, we found that the genetically engineered mouse model of K20R developed obvious hepatosplenomegaly, and expansion of bone marrows hematopoietic stem cells and downstream hematopoietic populations. We have examined homozygous EZH2 K20R mutant mice, as compared with the wildtype animals. We found that the xiphoid process of K20R mutant mice is more pronounced and harder than that of the wildtype mice (Figure7—figure supplement 2).

However, we could not clearly identify any other obvious morphological alterations in the K20R homozygous mutant mice, such as brain, heart, kidney, eye, and stomach, did not find clear morphology defects in these tissues. It is important to note that FDA has granted approval to tazemetostat, an EZH2 inhibitor, for follicular lymphoma. Our current studies suggest that the altered hematopoiesis in the K20R mutant mice indicate that the hematopoietic system may be the most vulnerable tissue to the K20R mutation in EZH2. A more detailed analysis is required to detect other molecular or cellular alterations in K20R mutant mice. We also examined hematopoietic tissues in LSD1^fl/fl^ conditional KO and L3MBTL3^fl/fl^ conditional KO mice using vav-iCre for hematopoietic tissues. The LSD1^fl/fl^ conditional KO with vav-iCre are embryonic lethal around E17.5-E18.5 and the KO fetal lives produced much reduced levels of red blood cells. The L3MBTL3^fl/fl^ conditional KO mice with vav-cre are alive, with the accumulation of EZH2, SUZ12, EED, and GFI1B in the spleens (4.5 months), as compared to the wild-type animals. These elevated protein patterns are quite similar to that of the K20R mutant mice in Figure 8E of the revision. Our additional analysis for hematopoietic alterations in the L3mbtl3 fl/fl conditional mice with the vav-iCre transgenic mice revealed the total increases of whole bone marrow (BM), LSK, and LK cells in the Vav-iCre;L3MBTL3fl/fl KO mice. This phenotype is similar to that of K20R mutant mice. However, we did this examination only once in one Vav-iCre;L3MBTL3fl/fl KO mouse so it is not statistically valid for publication. We will only show the result here but not in the revision. The homozygous Dcaf5 KO mice are alive and we are still breeding for more mutant mice to examine the possible alterations in the whole bone marrow (BM), LSK, and LK cells.We have also performed new experiments with quantitative RT-PCR of wildtype and K20R bone marrow samples to analyze the classical H3K27me3 repressed targets. Our evaluation of EZH2 repressed target genes revealed that some of H3K27me3 repressed target genes are downregulated, such as Strc, Syngap1, Bmi1, Ltgb1, Ppfi1a, and Runx3, derived from a recent paper (Blood, 138, 221-233, 2021) and the new data are included in new Figure 7—figure supplement 1D in the revision.

The upregulation of GFI1B in hematopoietic system is surprising but it is very reproducible. Since GFI1B is a transcriptional repressor, we need to know more about the induction of GFI1B by K20R mutation. Previous studies have shown that GFI1B, GATA1, and EZH2 physically interacted and cooperated to suppress target genes such as Hes1 promoter (MCB 32, p3624-3638, 2012; PNAS, 2014, E344-353; Mol. Cell 36, 682-695, 2009). GFI1B can bind to the GFI1/GFI1B sites close to its mRNA start site and can repress its own transcription (Nucleic Acid Res. 33, 987-998, 2005). However, in medulloblastoma, EZH2 inactivation upregulates Gfi1. Whether upregulated GFI1B transcription in the K20R mice allows EZH2 to stabilize its bound GFI1B, which is silenced in spleen but not in the bone marrow (Nucleic Acid Res. 33, 987-998, 2005), remains to be determined. Further investigation is required to address the regulation of K20R mutant of EZH2 on Gfi1b and to determine whether the Gfi1b gene is a direct EZH2 target.

Reviewer #1 (Recommendations for the authors):1. The overall model is that SET7-mediated EZH2K20 methylation promotes EZH2 protein degradation, which suggests that this mark negatively regulates EZH2. However, Figure 4F (and Figure 5D) shows that the EZH2 protein level does not change upon overexpression of SET7, and in contrast, the global H3K27m3 level increases, indicating an overall positive role. Although the authors attribute this to the decrease of EZH2 S21 phosphorylation, it seems paradoxical that a modification positively and negatively regulates the modified protein simultaneously. If the main function of K20me is to promote protein degradation, why do cells also utilize it to activate EZH2 enzymatic activity?

We appreciate the reviewer’s questions. We are sorry that the original Figure 4F showed that the ectopic SET7 expression in T47D cells did not significantly reduce EZH2 and that T47D cells are unusual because these cells express lower endogenous L3MBTL3 levels (new Figure 6, supplement 2A). Since L3MBTL3 is required for the degradation of methylated EZH2 protein, the low levels of L3MBTL3 in T47D cells may contribute to the defection in EZH2 degradation. The data in T47D cells are quite similar to that of L3mbtl3 KO in MEFs (Figure 6E) that EZH2 does not response to signals for degradation. Further evidence supporting that the L3MBTL3 level is critical for EZH2 degradation is when we ectopically expressed L3MBTL3 in T47D cells, these cells can promote EZH2 degradation in response to proteolytic signal (Figure 6D). In Figure 5D, SET7 was transiently expressed into 293T cells, usually only a fraction of cells are expressing SET7 in these type experiments. The total EZH2 levels may not change significantly if the fraction of cells that express SET7 is relatively limited to cause any detectable EZH2 changes.

2. The effect of LSD1 KO on EZH2 is drastic, leading to almost diminished levels of EZH2 and H3K27me3 (Figures 1, 2, 5f, etc), suggesting that the majority of EZH2 is methylated. Quantitative MS needs to be performed to assess the K20 methylation levels +/-LSD1 KO and +/- MG132. More importantly, based on the proposed model of K20me-S21phos crosstalk, one would expect an increase in the H3K27me3 level upon LSD1 KO, as seen above in SET7 overexpression.

We appreciate the reviewer’s constructive suggestions. In response, the LSD1 KO effects on EZH2 and H3K27me3 are most pronounced in the LSD1 conditional KO mice. The silencing of LSD1 also induced the proteolysis of EZH2 protein in MEFs, and other cells such as PA-1, H1299, HCT116, and G401 cells. However, T47D cells are unusual because L3MBTL3 appears to be in limiting in these cells for the methylation-dependent EZH2 proteolysis, as compared to other cells or cells in mice and MEFs (Figure 6 supplement 2A-2C, Figure 6A-6C). We found that LSD1 serves a demethylase that removes the methyl group from the methylated K20 in EZH2 and loss of LSD1 should increase the methylated EZH2 to be targeted by L3MBTL3 and CRL4^DCAF5^ ubiquitin ligase for proteolysis. Consistent with this hypothesis, our data showed that loss of LSD1 promotes EZH2 degradation and downregulates H3K27me3. In response to reviewer’s comment, we have conducted new experiments to use siRNAs to knockdown LSD1 and examine the levels of EZH2 and H3K27me3 in HCT116 cells. Our studies found that both EZH2 and H3K27me3 are downregulated by LSD1 silencing in HCT116 cells. We also directly measured the effects on EZH2-K20me after LSD1 silencing in HCT116 cells. We found that loss of LSD1 reduced the level of K20 methylation of EZH2, and the reduction of EZH2 can be rescued by the treatment of LSD1-deficient cells with the protease inhibitor MG132 (new Figure 4D). We also measured EZH2-K20me level in Lsd1^fl/fl^ conditional and L3MBTL3-knockout mice. As shown in Figure 4E, LSD1 deletion in Lsd1 ^fl/fl^ mice using Nestin-Cre decreased the methylation of EZH2 at K20. Conversely, L3MBTL3 knockout resulted in the increased EZH2 K20me (Figure 4F).

In response to reviewer’s comments on using the quantitative MS to assess the K20 methylation levels +/-LSD1 KO and +/- MG132. We have tried several rounds of mass spectrometry mediated analysis to detect the methylated K20 or the phosphorylated S21 in EZH2 using mouse single LSD1^fl/fl^ and/or L3mbtl3^fl/fl^ mice, or the combination of double knockout of Lsd1^fl/fl^ and L3mbtl3^fl/fl^ mice with different types of transgenic cre recombinase mice. However, unfortunately, we could not obtain the chymotryptic peptides containing K20/S21, using between 1-10 mouse knockdown embryos, as mentioned above in response to main question #2. If methylated K20 and/or phosphorylated S21 exist in a fraction of total EZH2 protein population, it seems that the requested quantitative MS is technically very difficult for us to measure the levels of methylated K20 and/or phosphorylated S21 in EZH2 since the detections require at least the full coverage of all EZH2 chymotryptic peptides.

3. Figure 5F, it is surprising that S21A leads to global loss of H3K27me3, given that the endogenous EZH2 still exists presumably. Endogenous EZH2 and EZH2K20me need to be probed. More importantly, is S21A enzymatically dead and does it function as a dominant negative mutant?

We appreciate the reviewer’s comments. We have answered the similar main question #4 above and we will respond similarly here: In Figure 5F, the ectopic expression of Flag-tagged EZH2-S21A mutant did not change the levels of Flag-tagged EZH2 or endogenous EZH2, and the levels of H3K27me3. But when LSD1 is silenced, both EZH2 and H3K27me3 decreased to similar levels in cells expressing Flag-EZH2 wildtype and Flag-S21A mutant (Figure 5F). We agree that EZH2-S21A mutant is not enzymatically dead, but our data showed that the EZH2 S21A protein is still sensitive to LSD1 silencing. As to the endogenous EZH2 and EZH2K20me, we did new experiments which showed that endogenous EZH2 and methylated K20 form of EZH2 were downregulated after LSD1 silencing in cells expressing Flag-EZH2 wildtype, K20R, and S21A proteins.

4. Figure 5G and H, overexpression of L3MBTL3 and DCAF5 promotes EZH2 ubiquitination. But why it does not affect global EZH2 levels?

We thank the reviewer’s comment. In Figure 5G and 5H, these are transiently transfected 293T cells so a fraction of these cells are transfected. In these experiments, we tried to use proteasome inhibitor MG132 (5 μg/ml) to treat the transfected cells for last 6 hours to stabilize the polyubiquitinated EZH2. Therefore, in these experiments, we can detect the polyubiquitinated EZH2, but the total levels of EZH2 in the transfected cells are not significantly affected. To address the reviewer’s comment, separate and new experiments were conducted, involving the establishment of stably and ectopically expressed and Flag-tagged L3MBTL3 in HCT116 cells and we observed that overexpression of L3MBTL3 can decrease the protein level of EZH2 (Figure 2—figure supplement 2B). In the revision, we have added the “proteasome inhibitor MG132 (5 μg/ml) was added for last 6 hours.” In the Figure 5 legend.

5. Figure 6, the authors propose that L3MBTL3 levels in cells determine the outcome of EZH2K20 methylation. if this is the case, one would expect to see a negative correlation of EZH2 and L3MBTL3 at the protein level across cell lines.

We appreciate the reviewer’s excellent comments. We checked the protein levels of EZH2 and L3MBTL3 in the cell lines we used. It seems to be the case in T47D cells that there is a negative correlation of EZH2 and L3MBTL3 at the protein levels. Unfortunately, we did not see a direct correlation between EZH2 and L3MBTL3 in other cells. It is possible that L3MBTL3 is one of the limiting factors that are altered in cancer cells and T47D is the only cell line we have to show this type of correlation at protein levels and the functional defect in methylated EZH2 proteolysis.

Reviewer #2 (Recommendations for the authors):1. P7, the conclusion for the section "Deletion of mouse L3mbtl3 gene causes the accumulation of EZH2 protein" is inappropriate. The data presented in Figure 2 only demonstrated that L3mbtl3 deletion can rescue LSD1 silencing reduced EZH2 protein levels, unrelated to L3MBTL3-dependent proteolysis.

We thank the reviewer for the comments. We are aware of that L3MBLT3 deletion sometimes causes increased levels of EZH2 protein, whereas loss of L3MBTL3 only rescued the downregulation of EZH2 after LSD1 loss. For example, in Figure 2 we presented the Western Blots for EZH2 protein levels in the mouse L3mbtl3 KO and WT embryos (Figure 2A), the Western blots of L3mbtl3 KO and WT mouse embryo derived MEFs (Figure 2B), the Western blots of nestin-Cre mediated deletion of conditional L3mbtl3 fl/fl and not deleted L3mbl3^fl/+^MEFs (Figure 2C), and immunostaining of EZH2 protein in the embryonic brain (Figure 2D). These data showed the elevated mouse EZH2 protein levels in the L3mbtl3 KO or conditional KO mouse cells, no LSD1 deletion in these samples. On the other, we also showed L3MBTL3 loss can rescue the EZH2 protein levels in LSD1 deleted MEFs. For example, induced L3mbtl3 deletion can rescue the downregulated EZH2 protein levels in the induced Lsd1 deletion MEFs (Figure 2E) and in cultured cells (Figure 2F and 2G). We are still investigating why loss of L3MBTL3 can sometimes elevate EZH2 protein in some cells (Figure 2G) but in other cases L3mbtl3 deletion can only rescue the downregulated EZH2 levels in LSD1 deficient cells (Figure 2E and 2F). It is possible the fraction of EZH2 methylation, siRNA efficiency, and cell proliferation may affect these outcomes.

In response reviewers’ comments, we replaced the sentence of “Deletion of mouse L3mbtl3 gene causes the accumulation of EZH2 protein” with the new one: “L3MBTL3 regulates the stability of EZH2 protein” in the revision. In addition, we also used CRISPR/Cas9 editing to generate a L3mbtl3-knockout HCT116 cell line, in which the expression of EZH2 protein was higher than that in L3MBTL3-wildtype cell (Figure 2—figure supplement 2A). To further evaluate the effect of L3MBTL3 on EZH2, we ectopically and stably expressed Flag-tagged L3MBTL3 in HCT116 cells. As shown in Figure 2—figure supplement 2B, EZH2 protein level decreased when L3MBTL3 was overexpressed in HCT116 cells.

2. Figure 4 needs to be reorganized to display more logically.

We appreciate the comments of the reviewer. In response, we have reorganized the panels in Figure 4 in the revision to show the primary loss effects of LSD1, a demethylase, are mediated through the SET7 methyltransferase and L3MBTL3. EZH2 contains a conserved SET7 methylation motif around K20 (Figure 4A and 4B), and LSD1 can demethylate the methylated K20 in vitro (Figure 4C). We have conducted new experiments showing that loss of LSD1 in HCT116 cells can destabilize EZH2 protein (Figure 4D), and conditional deletion of LSD1 causes the downregulation of K20 methylated EZH2 in the mouse, whereas loss of L3mbtl3 in the mouse increases the K20-methyleted form of EZH2 (Figure 4F). Ectopic expression of SET7 can downregulate EZH2 protein and increase K20 methylation (Figure 4G and 4H). The silencing of SET7 can rescue the downregulation of EZH2 in LSD1 silenced cells (Figure 4I, Figure 4 supplement 2A and 2D). We have also added new experiments to show that PHF20L1 prevents EZH2 degradation (Figure 4 supplement 2B and 2C). We hope our new arrangement for Figure 4 the revision is better than the original version.

3. Figure 6 is the main evidence to support one of the important conclusions that the methylation-phosphorylation switch regulates the stability of EZH2, but only the K20me and S21p of EZH2 in MEFs were presented in Figure 6B. How about the K20me and S21p levels in T47D and H1299 cells after MK2206 treatment?

We thank the reviewer for the helpful comments. In response, we have conducted new experiments to show that in addition to MEFs in Figure 6B, AKT inhibitor MK2206 inhibited S21 phosphorylation, increased the K20-methylation, and reduced EZH2 protein levels in human teratocarcinoma PA-1 cells and lung carcinoma H1299 cells (new Figure 6C, Figure 6 supplement 2C). In T47D cells, MK2206 also increased K20me, with decreased S21p levels of EZH2 (Figure6-supplement 2B). However, MK2206 does not significantly induce EZH2 proteolysis due to the low level of L3MBTL3 (Figure6-supplement 2A) in the revision.

4. It seems that the low expression and alterations of L3MBTL3 are not rare in cancer cells. Further discussion is needed about how broadly the methylation-phosphorylation switch controls EZH2 stability during development and disease initiation/progression.

We thank the reviewer for the constructive comments. In response, we have so far found that T47D cells does not significantly cause EZH2 proteolysis after LSD1 silencing or MK2206 treatment, likely due to the low level of L3MBTL3 since expression of L3MBTL3 can restore the degradation activity of EZH2 in T47D after MK2206 treatment (Figure 6D). We will conduct systematic analysis in various cancer cells to characterize the response of LSD1 silence in the future, as this is likely a very important alteration in certain cancer cells.

Reviewer #3 (Recommendations for the authors):1. There are no data directly demonstrating that enzymatic activity dead mutants of LSD1, SET7, and L3MBTL3 lose their roles in the regulation of EZH2 methylation and protein stability. Almost all the experiments utilized knockdown or knockout strategies, which cannot exclude the off-target or secondary effects. Rescue experiments may be essential.

We thank the reviewer for the positive comments for our manuscript and the specific and important questions for the direct enzymatic evidence of LSD1, SET7, and L3MBTL3. In response, we have conducted new experiments to silence endogenous LSD1 by using the siRNA against the 3’UTR region (si-Lsd1-3’UTR) that led to the downregulation of EZH2 and the induced proteolysis of EZH2 but this effect can be rescued by the ectopic expression of a functional wildtype Flag-LSD1 cDNA that does not contain the 3’-UTR, but not by the expression of a catalytically dead mutant LSD1 containing only the amino-terminal 1-531 amino acid residues of the Flag-LSD1 cDNA (new Figure 1-supplement 1B and 1C). These experiments indicate that the wildtype LSD1 is functional to suppress the si-Lsd1-3’UTR effects of the endogenous LSD1 on EZH2 protein. We also conducted new experiments using MK2206 treatment in PA-1, T47D, and H1299 cells to show that inhibition of AKT and S21 phosphorylation leads to the increase methylation of K20 in EZH2. In new Figure 5D, we showed that expression of wildtype SET7, but not its catalytically inactive form of mutated SET7 (the H297A mutant), can methylate EZH2 and promote the binding of EZH2 to L3MBTL3. Our data further shoed that expression of L3MBTL3 is sufficient to restore the ability of T47D cells to reduce EZH2 protein in response to MK2206 (Figure 5D). Together, these data in the revised manuscript indicate that LSD1,SET7, and L3MBTL3 are enzymatically or functionally to act as demethylase, methyltransferase, or methyl lysine binding proteins for the dynamic and enzymatic regulation of EZH2 protein.

2. In Figure 1F, the authors showed that LSD1 knockdown can indeed downregulate the protein levels of EZH2 in T47D. But if L3 is low in T47D, shouldn't they see no effect of LSD1 on EZH2 protein stability?

We greatly appreciate and thank the reviewer’s excellent comments. In response, T47D cells are quite unique in their response to LSD1 silencing or MK2206 treatment. Unlike other cells, the L3MBTL3 level is significantly low in T47D cells, and EZH2 protein is usually quite resistant to LSD1 silencing or MK2206 treatment (Figure 6A and also see Science, 310, 306-310, 2005). Our studies showed that ectopic expression of L3MBTL3 is sufficient to restore the ability to proteolyze EZH2 in response to MK2206 in T47D cells (Figure 6D), indicating L3MBTL3 is limiting in these cells. Occasionally, however, we can detect that silencing of LSD1 can reduce EZH2 protein levels in T47D cells, probably due to the low cell density during the siRNA experiment. However, this EZH2 response to LSD1 silencing is very hard to observe frequently. We apologize that we put this result in the original Figure 1F as we did not realize the problem in T47D cells for EZH2 proteolysis. On the other hand, our data in MEFs showed that MK2206 caused EZH2 downregulation (Figure 6B). For the revision, we have conducted new experiments in other cells such as teratocarcinoma PA-1 cells, lung carcinoma H520, or H1299 cells to show that LSD1 silencing caused proteolysis of EZH2 (Figure 6C, Figure 6 figure supplement 2B and 2C). We have used these data to replace the original Figure 1F T47D data with our new data for PA-1 and H520 cells (new Figure 1F and Figure1—figure supplement 1A). We thank again for reviewer’s excellent and helpful comments on our data in T47D cells.

3. In Figures 2D and 3D, shouldn't we see a global increase in the staining intensities of EZH2 and H3K27me3? It is not sure why the authors highlighted some regions of the coronal sections. If they think only in those regions EZH2 and H3K27me3 levels were changed, please explain the specificity of those regions.

We thank the reviewer for the question. In response, we did observe a general increase of EZH2 signal in or immunostaining of L3mbtl3 or Dcaf5 deleted brain tissues, as shown in Figures 2D and 3D. We highlighted the cortical and ventricular zones of the neocortex in the mouse embryonic brains at E1.5 to illustrate the accumulation of EZH2 protein, which are also associated with high levels of H3K27me3 (Figure 2D and Figure 3D). During embryonic development at E15.5, the proliferation rates vary in different regions of the embryonic brain. The highly stained EZH2 regions are likely to be associated high proliferation rates during mouse brain embryonic development at E15.5, as it is well known that EZH2 expression is associated with proliferation.

3. Does the antibody recognize di- or tri-methylation of EZH2 at K20 or mono-methylation at other lysine residues? These types of modified peptides may be included in the in vitro dot assay to further prove the specificity of the antibody. It is also unclear if SET7 is the major methyltransferase responsible for the methylation of EZH2 at K20 in cells.

We appreciate the excellent comment from the reviewer. In response, we have conducted new dot blot analyses of our anti-K20 methylation antibodies for their ability to recognize the EZH2-K17, EZH2-K17me1, EZH2-K20me2, and EZH2-K20me3 peptides. Our results showed that EZH2-K20me antibody also recognize di- and tri-methylated EZH2 at K20, but not mono-methylated EZH2 at K17 (Figure 4-supplement 1). However, SET7 was isolated to mono-methyate H3K4, it is likely that K20 in EZH2 is mono-methylated. We also tried several times to use mass spectrometry proteomics by isolating EZH2 proteins from the conditional single KO of Lsd1^fl/fl^ or L3mbtl3^fl/fl^, and also the double KO of Lsd1^fl/fl^/L3mbtl3^fl/fl^ embryos by breeding the conditional knockout mice of Lsd1^fl/fl^, L3mbtl3^fl/fl^, and their combined knockout mice, to analyze K20 methylation. However, we found that even up to 10 homozygous conditional knockout mouse tissues, the chymotryptic K20 peptides were not identified by the Orbitrap mass spectrometry (please see above for response to main question #2). We feel that it is quite technically difficult to use MS to detect EZH2 K20 methylation or S21 phosphorylation to address this question at this moment, as we have to use chymotrypsin, which is less frequently used in MS, and also we need more than the full coverage of all EZH2 chymotryptic peptides for this purpose (depending also on the fraction of methylated or phosphorylated EZH2 peptides in our samples).

4. In Figure 4D-F, shouldn't manipulation of SET7 change the protein levels of EZH2? Overexpression of SET7 leads to downregulation of EZH2 (but H3K27me3 in Figure 4F was actually upregulated), whereas knockdown of SET7 should stabilize EZH2.

We thank the reviewer’s excellent questions. The original Figure 4D-4F were conducted in H1299 cells stably expressing the Flag-tagged EZH2. In these cells, we showed that siRNA-mediated silencing of LSD1 caused the reduction of the Flag-EZH2 protein, whereas co-silencing of LSD1 and SET7 prevented the reduction of Flag-EZH2 in LSD1 deficient cells. The key question for the data here is why the EZH2 protein level did not increase when SET7 is silenced? Since most of our repeated silencing experiments for SET7 or L3MBTL3 in cultured H1299 cells only showed that they restabilized EZH2 proteins reduced after the silencing of LSD1, we would suggest that only a fraction of EZH2 is K20 methylated and the methylated EZH2 may also bind to other proteins, such as PHF20L1, which we previously found that PHF20L1 binds to the methylated K42 of *Sox2* (JBC, 294 (2), 476-489, 2019) to prevent the degradation of methylated *Sox2* by L3MBTL3 and CRL4^DCAF5^-mediated proteolysis. In response to reviewer comments, we have examined the effects of silencing PHF20L1 on EZH2 and found that similar to *Sox2*, loss of PHF20L1 caused the downregulation of EZH2, which can be rescued by L3MBTL3 silencing (new Figure 4—figure supplement 2B and 2C in revision). In addition, we have conducted new experiments showing that ectopic expression of SET7 in human colorectal carcinoma HCT116 cells led to the reduction of both EZH2 protein and H3K27me3 (new Figure 4G in revision). Notably, we usually found that the deletion effects of LSD1 and L3MBTL3 are most pronounced in mouse embryos and in mouse embryonic stem cells (JBC, 294 (2), 476-489, 2019, and Nature Communications volume 13, Article number: 6696, 2022, https://www.nature.com/articles/s41467-022-34348-9) but less effective in many cultured cancer cell lines. While we are still investigating the mechanistic difference between embryonic stem cells and somatic cancer cells, it seems that the fraction of methylated K20 protein is relatively small in lung carcinoma H1299 cells so that the loss of SET7 did not often induce elevated EZH2 protein levels. One issue with H1299 cells is that they also express relatively low level of L3MBTL3 (Figure 6, figure supplement 2A) and we don’t know whether it affects the effect of L3MBTL3 silencing. In the original Figure 4F, ectopic expression of SET7can induce K20 methylation. However, due to the low level of L3MBTL3 in T47D cells (Figure 6 supplement 6A), the increased levels of K20 methylation did not cause the downregulation of EZH2 protein.

5. In Figure 5B, there is no data showing the "gradual reduction of the S21-phosphorylated form of EZH2". It is interesting to see that phosphorylated AKT was actually reduced during mouse embryonic development. Why is that?

We thank the reviewer for this comment. In response, we apologize for statement of the “gradual reduction of S21-phorylated form of EZH2”in the original manuscript. Instead, we should use “gradual reduction”to describe the AKT phosphorylated form during embryonic development. However, we don’t know why AKT1 phosphorylation is downregulated during development. To avoid any confusion, we have removed this part in the revision. On the other hand, we have measured the S21p of EZH2 several times during the mouse embryonic developmental process and obtained the similar results. We usually could only detect a very faint S21-phosphorylarion band during development, indicating the low levels of EZH2-S21p from E14 before birth. However, K20 methylation and S21 phosphorylation both increase right after animal birth (P0), associated with the reduction levels of EZH2 proteins and H3K27me3, suggesting that EZH2 is likely associated with proliferation during embryonic stages, and both activity and proteolysis of EZH2 increase after birth. In the revision, we have presented our data in the updated Figure 5A.

6. The converged effect of EZH2 K20 methylation and S21 phosphorylation on H3K27me3 is confusing. K20 methylation of EZH2 destabilizes the histone methyltransferase, whereas S21 phosphorylation impairs its enzymatic activity. However, K20 mono-methylation prevents S21 phosphorylation. Then which modification will win over in terms of deciding the H3K27me3 levels? And why?

This is an important question and we thank the reviewer for the comment. In response, we believe that EZH2 exists in several forms and not all fractions of EZH2 protein are methylated or phosphorylated. Therefore, the fraction of EZH2 protein that is not modified by K20-methylation or S21-phosphorylation is likely catalytically active for H3K27 trimethylation. The methylated K20 fraction of EZH2 is also protected by PHF20L1, as our studies showed the loss of PHF20L1 destabilizes EZH2 protein (new figure4, figure supplement 2B and 2C). We previously found that PHF20L1 binds to the methylated K42 of *Sox2* (JBC, 294 (2), 476-489, 2019) to prevent the degradation of methylated *Sox2* by L3MBTL3 and CRL4^DCAF5^-mediated proteolysis. Since the K20 methylation motif is very similar to that of K42 motif in *Sox2*, we think it is likely that the PHF20L1-protetected K20-methylated EZH2 fraction may also be catalytically active. We have added these possibilities in the discussion of the revision.

7. The conflicting result that the methylation-phosphorylation switch of EZH2 is defective in T47D cells is very confusing. If this is due to the low level of L3MBTL3, why LSD1 knockdown in T47D can still downregulate EZH2 protein then (Figure 1)? Although the authors showed in Supplemental Figure 2 that L3MBTL3 levels are various in different cell lines, they didn't really show whether in those L3MBTL3-low cells other than T47D, they see the same results as in T47D.

We appreciate the comments of the reviewer. In response, we found the levels of L3MBTL3 are quite variable in different cancer cell lines. EZH2 in T47D cells is very unusual and is quite resistant to LSD1 silencing, possibly due to the low level of L3MBTL3 in T47D cells, as mentioned in response to question #2 above.MK2206 usually did not produce significant effects on EZH2 protein in T47D cells and only occasionally downregulates EZH2 when LSD1 is silenced. To avoid confusion, we have replaced T47D results in Figure 1F with that of PA-1 cells in the revision, as mentioned above in response to question #2. Since the reduction of EZH2 in some other cells, such as H1299 cells with low L3MBTL3 levels, can repeatedly observed after LSD1 silencing, it is likely that the regulation of EZH2 proteolysis is quite complicated so that other activities may promote the degradation of EZH2 even when L3MBTL3 levels are relatively low. While expression of L3MBTL3 in T47D cells is sufficient to reproducibly promote EZH2 degradation after MK2206 treatment (Figure 6D), further systematic analyses are required to determine the factors that involved in EZH2 proteolytic degradation and how these processes are altered in various cancer cells.

8. The most direct data to demonstrate the negative feedback between K20 methylation and S21 phosphorylation is to detect the levels of these two modifications in cells expressing wild-type EZH2 or K20R or S21A mutant using the specific antibodies detecting the modified EZH2. They can also overexpress SET7 or AKT when they express the corresponding mutant.

We thank the reviewer for the constructive suggestions. In response, we have conducted new experiments in PA-1 (new Figure 6C) and H1299 cells (new Figure 6 supplement 2C) to show that increased K20 methylation and reduction of S21-phosphorylation occur when these cells are responding to MK2206, similar to that of MEFs in Figure 6B. T47D cells can response to MK2206 by increasing K20 methylation and reducting S21-phosphorylation (new Figure 6 supplement 2B). However, MK2206 usually does not induce the reduction of EZH2 protein in T47D cells. We hope the new data in the revision help to address the methylation-phosphorylation switch that regulates EZH2 protein levels and activities.

9. It seems that the methylation-phosphorylation switch of EZH2 is not specific to cancer (observed in MEF) nor any specific types of cancer. Why does K20R overexpression only induce hematopoiesis in the GEMM model? Did the authors see similar results in Lsd1/L3mbtl3/Dcaf5-knockout mice?

We thank the reviewer for this excellent comment. In response, we have examined other regions/organs in homozygous EZH2 K20R mutant mice, as compared with the wildtype animals. We found that the xiphoid process of K20R mutant mice is more pronounced and harder than that of the wildtype mice (Figure7—figure supplement 3). However, we could not find any other obvious morphological alteration in the K20R homozygous mutant mice. It is interesting that FDA granted approval to tazemetostat, an EZH2 inhibitor, for follicular lymphoma. Our current studies suggest that K20R mutant mice display the altered hematopoiesis may indicate the hematopoietic system may be most vulnerable to K20R mutation in EZH2. A more detailed analysis is required to detect other molecular or cellular alterations in K20R mutants. We have conducted analysis for hematopoietic alterations in the L3mbtl3^fl/fl^ conditional mice with the vav-iCre transgenic mice for hematopoiesis. We have examined whole bone marrow, LSK, and LK populations in Vav-iCre;L3MBTL3^fl/fl^ mice and observed the total increase of whole BM, LSK, and LK cells, and this phenotype is similar to that of K20R mutant. However, we did this experiment only once in one L3mbtk3 KO mouse, so it is not statistically valid for publication. We would only show theL3mbtl3 KO result here but not in the revision.

10. If H3K27me3 is increased in K20R-expressing mice, why is GFI1B expression upregulated? Is it an H3K27me3-independent function of EZH2? How will the authors reconcile the increase in H3K27me3 levels in K20R-expressing MEF with cytoplasmic localization of this mutant form of EZH2 (Supplemental Figure 4)?

We thank the reviewer for the comments. In response, the upregulation of GFI1B in hematopoietic system is likely regulated by multiple processes. Previous studies have shown that GFI1B, GATA1, and EZH2 physically interacted and cooperated to suppress target genes such as Hes1 promoter (MCB 32, p3624-3638, 2012; PNAS, 2014, E344-353; Mol. Cell 36, 682-695, 2009). GFI1B is a transcription repressor that can bind to the GFI1/GFI1B sites close to its mRNA start site and can repress its own transcription (Nucleic Acid Res. 33, 987-998, 2005). In medulloblastoma, EZH2 inactivation upregulates Gfi1. Whether upregulated GFI1B transcription in the K20R mice may allow EZH2 to stabilize its bound GFI1B, which is silenced in spleen but not in the bone marrow (Nucleic Acid Res. 33, 987-998, 2005). The upregulation of Gfi1b/GFI1B in K20R animals are highly reproducible. Further investigation is required to address the regulation of Gfi1b by the K20R mutant of EZH2.

Our studies suggest that K20R mutation stabilized EZH2 protein. However, a small fraction of K20R mutant protein is in the cytoplasm. It is likely that the nuclear K20R form of EZH2 remains active to trimethylate H3K27 in the nucleus. Further studies is required to determine the role of cytoplasmic EZH2 in the regulation hematopoietic system.

11. What about the classical genes that are repressed by H3K27me3? Are they downregulated in K20R-expressing cells?

We thank the reviewer for the suggestion. In response, we performed new RT-PCR analyses of wildtype and K20R bone marrow samples. We evaluated previously reported EZH2 repressed target genes (Blood 138, 221-233, 2021) and found that some of them are downregulated, such as Strc, Syngap1, Bmi1, Ltgb1, Ppfi1a, and Runx3. These new data are added in new Figure 7 supplement 1D in the revision.